# SCALE-INVARIANT BAYESIAN NEURAL NETWORKS WITH CONNECTIVITY TANGENT KERNEL

**Sung-Yub Kim[1], Sihwan Park[1], Kyungsu Kim[3,4,5],\* Eunho Yang[1,2]\***
Korea Advanced Institute of Science and Technology (KAIST)[1], AITRICS[2],
Samsung Medical AI Research Center[3], Sungkyunkwan University School of Medicine[4],
Massachusetts General Hospital and Harvard Medical School[5]
sungyub.kim@kaist.ac.kr, sihwan.park@mli.kaist.ac.kr
kskim.doc@gmail.com, eunhoy@kaist.ac.kr

## ABSTRACT

Studying the loss landscapes of neural networks is critical to identifying generalizations and avoiding overconfident predictions. Flatness, which measures the perturbation resilience of pre-trained parameters for loss values, is widely acknowledged as an essential predictor of generalization. While the concept of flatness has been formalized as a PAC-Bayes bound, it has been observed that the generalization bounds can vary arbitrarily depending on the scale of the model parameters. Despite previous attempts to address this issue, generalization bounds remain vulnerable to function-preserving scaling transformations or are limited to impractical network structures. In this paper, we introduce new PAC-Bayes prior and posterior distributions invariant to scaling transformations, achieved through the *decomposition of perturbations into scale and connectivity components*. In this way, this approach expands the range of networks to which the resulting generalization bound can be applied, including those with practical transformations such as weight decay with batch normalization. Moreover, we demonstrate that scale-dependency issues of flatness can adversely affect the uncertainty calibration of Laplace approximation, and we propose a solution using our invariant posterior. Our proposed invariant posterior allows for effective measurement of flatness and calibration with low complexity while remaining invariant to practical parameter transformations, also applying it as a reliable predictor of neural network generalization.

## 1 INTRODUCTION

Neural networks (NNs) have succeeded tremendously, but understanding their generalization mechanism in real-world scenarios remains challenging (Kendall & Gal, 2017; Ovadia et al., 2019). Although it is widely recognized that NNs naturally generalize well and avoid overfitting, the underlying reasons are not well understood (Neyshabur et al., 2015b; Zhang et al., 2017; Arora et al., 2018). Recent studies on the loss landscapes of NNs attempt to address these issues. For example, Hochreiter & Schmidhuber (1995) proposed the flat minima (FM) hypothesis, which states that loss stability for parameter perturbations positively correlates with network generalizability, as empirically demonstrated by Jiang et al. (2020).

However, the FM hypothesis still has limitations. According to Dinh et al. (2017), rescaling two successive layers can arbitrarily degrade a flatness measure while maintaining the generalizability of NNs. Meanwhile, Li et al. (2018) argued that weight decay (WD) leads to a contradiction of the FM hypothesis in practice: Although WD sharpens pre-trained NNs (i.e., decreased loss resilience), it generally improves the generalization. In short, they suggest that transformations on network parameters (e.g., re-scaling layers and WD) may lead to contradictions to the FM hypothesis. A thorough discussion on this can be found in Appendix E.

To resolve this contradiction, we investigate PAC-Bayesian prior and posterior distributions to derive a new scale-invariant generalization bound. As a result, our bound guarantees invariance for a general

---

*Correspondence to

class of function-preserving scale transformations with a broad class of networks. Specifically, our bound is more general than existing works (Tsuzuku et al., 2020; Kwon et al., 2021), both in terms of transformations (e.g., activation-wise rescaling (Neyshabur et al., 2015a) and WD with batch normalization (BN; Ioffe & Szegedy (2015))) that guarantee invariance and in terms of NN architectures. Therefore, our bound ensures no FM contradiction for the first time, which should not occur in practical NNs, including ResNet (He et al., 2016) and Transformer (Vaswani et al., 2017).

Our generalization bound is derived from scale invariances of prior and posterior distributions, guaranteeing not only its scale invariance but also the scale invariance of its substance, the Kullback-Leibler (KL) divergence-based kernel. We call this kernel an empirical Connectivity Tangent Kernel (CTK), as a modification of empirical Neural Tangent Kernel (Jacot et al., 2018) with the scale-invariance property. Moreover, we define a new sharpness metric as the trace of CTK, named **Connectivity Sharpness (CS)**. We show via empirical studies that CS predicts NN generalization performance better than existing sharpness measures (Liang et al., 2019; Neyshabur et al., 2017).

In Bayesian NN regimes, we connect the contradictions of the FM hypothesis with the issue of amplifying predictive uncertainty. Then, we alleviate this issue by using a Bayesian NN based on the posterior distribution of our PAC-Bayesian analysis. We name this Bayesian NN as **Connectivity Laplace (CL)**, as it can be seen as a variation of Laplace approximation (LA; MacKay (1992)) using a different Jacobian. Specifically, we demonstrate the major pitfalls of WD with BN in LA and show how to remedy this issue using CL.[1] We summarize our contributions as follows:

- Our novel PAC-Bayes generalization bound guarantees invariance for general function-preserving scale transformations with a broad class of networks (Sec. 2.2 and 2.3). We empirically verify this bound gives non-vacuous results for ResNet with 11M parameters (Sec. 2.4).

- Based on our bound, we propose a low-complexity sharpness metric CS (Sec. 2.5), which empirically shows a stronger correlation with generalization error than other metrics (Sec. 4.1).

- To prevent overconfident predictions, we show how our scale-invariant Bayesian NN can be used to solve pitfalls of WD with BNs, proving its practicality (Sec. 3 and 4.2).

## 2 PAC-BAYES BOUND WITH SCALE-INVARIANCE

This section introduces a data-dependent PAC-Bayes generalization bound without scale-dependency issues. To this end, we introduce our setup in Sec. 2.1, construct the scale-invariant PAC-Bayes prior and posterior in Sec. 2.2, and present the detailed bound in Sec. 2.3. Then, we demonstrate the effectiveness of this bound for ResNet-18 with CIFAR in Sec. 2.4. An efficient proxy of this bound without complex hyper-parameter optimization is provided in Sec. 2.5.

### 2.1 BACKGROUND

**Setup and Definitions.** We consider a Neural Network (NN), $f(\cdot, \cdot) : \mathbb{R}^D \times \mathbb{R}^P \to \mathbb{R}^K$, given input $x \in \mathbb{R}^D$ and network parameter $\theta \in \mathbb{R}^P$. Hereafter, for simplicity, we consider vectors as single-column matrices unless otherwise stated. We use the output of NN $f(x, \theta)$ as a prediction for input $x$. Let $\mathcal{S} := \{(x_n, y_n)\}_{n=1}^N$ denote the independently and identically distributed (i.i.d.) training data drawn from true data distribution $\mathcal{D}$, where $x_n \in \mathbb{R}^D$ and $y_n \in \mathbb{R}^K$ are input and output representations of $n$-th training instance, respectively. For simplicity, we denote concatenated input and output of all instances as $\mathcal{X} \in \mathbb{R}^{ND}$ and $\mathcal{Y} \in \mathbb{R}^{NK}$, respectively, and $f(\mathcal{X}, \theta) \in \mathbb{R}^{NK}$ as a concatenation of $\{f(x_n, \theta)\}_{n=1}^N$. Given a prior distribution of network parameters $p(\theta)$ and a likelihood function $p(\mathcal{S}|\theta) := \prod_{n=1}^N p(y_n|f(x_n, \theta))$, Bayesian inference defines posterior distribution of network parameter $\theta$ as $p(\theta|\mathcal{S}) := \exp(-\mathcal{L}(\mathcal{S}, \theta))/Z(\mathcal{S})$, where $\mathcal{L}(\mathcal{S}, \theta) := -\log p(\theta) - \sum_{n=1}^N \log p(y_n|x_n, \theta)$ is training loss and $Z(\mathcal{S}) := \int p(\theta)p(\mathcal{S}|\theta)d\theta$ is the normalizing factor. For example, the likelihood function for a regression task will be Gaussian: $p(y|x, \theta) = \mathcal{N}(y|f(x, \theta), \sigma^2 \mathbf{I}_K)$ where $\sigma$ is (homoscedastic) observation noise scale. For a classification task, we treat it as a one-hot regression task following Lee et al. (2019a); He et al. (2020). While we adopt this modification for theoretical tractability, Hui & Belkin (2021) showed this modification offers good performance competitive to the cross-entropy loss. Details on this modification are given in Appendix C.

---

[1] https://github.com/sungyubkim/connectivity-tangent-kernel

**Laplace approximation.** In general, the exact computation for the Bayesian posterior of a network parameter is intractable. The Laplace approximation (LA; MacKay (1992)) is proposed to approximate the posterior distribution with a Gaussian distribution defined as $p_{\text{LA}}(\psi|\mathcal{S}) \sim \mathcal{N}(\psi|\theta^*, (\nabla^2_\theta \mathcal{L}(\mathcal{S}, \theta^*))^{-1})$ where $\theta^* \in \mathbb{R}^P$ is a pre-trained parameter with training loss and $\nabla^2_\theta \mathcal{L}(\mathcal{S}, \theta^*) \in \mathbb{R}^{P \times P}$ is a Hessian matrix of loss function w.r.t. parameter at $\theta^*$.

Recent works on LA replace the Hessian matrix with (Generalized) Gauss-Newton matrix to make computation easier (Khan et al., 2019; Immer et al., 2021). With this approximation, the LA posterior of the regression problem can be represented as

$$p_{\text{LA}}(\psi|\mathcal{S}) \sim \mathcal{N}(\psi|\theta^*, (\underbrace{\mathbf{I}_P/\alpha^2}_{\text{Damping}} + \underbrace{\mathbf{J}_\theta^\top \mathbf{J}_\theta/\sigma^2}_{\text{Curvature}})^{-1}) \tag{1}$$

where $\alpha, \sigma > 0$, $\mathbf{I}_P \in \mathbb{R}^{P \times P}$ is a identity matrix, and $\mathbf{J}_\theta \in \mathbb{R}^{NK \times P}$ is a concatenation of $\mathbf{J}_\theta(x, \theta^*) \in \mathbb{R}^{K \times P}$ (Jacobian of $f$ w.r.t. $\theta$ at input $x$ and parameter $\theta^*$) along training input $\mathcal{X}$. Inference with LA requires a further sub-curvature approximation for modern NN architectures (e.g., ResNet (He et al., 2016) and Transformer (Vaswani et al., 2017)) because of the prohibitively large covariance matrix. This approximation includes diagonal, Kronecker-factored approximate curvature (KFAC), last-layer, and subnetwork approximation (Ritter et al., 2018; Kristiadi et al., 2020; Daxberger et al., 2021). Meanwhile, it is well known that proper selection of prior scale $\alpha$ is needed to balance the dilemma between overconfidence and underfitting in LA.

**PAC-Bayes bound with data-dependent prior.** We consider a PAC-Bayes generalization error bound of classification task used in McAllester (1999); Perez-Ortiz et al. (2021) (especially equation (7) of Perez-Ortiz et al. (2021)). Let $\mathbb{P}$ be a PAC-Bayes prior distribution over $\mathbb{R}^P$ independent of training dataset $\mathcal{S}$, and $\text{err}(\cdot, \cdot) : \mathbb{R}^{K \times K} \to [0, 1]$ be an error function defined separately from the loss function. For any constant $\delta \in (0, 1]$ and $\lambda > 0$, and any PAC-Bayes posterior distribution $\mathbb{Q}$ over $\mathbb{R}^P$, the following holds with probability at least $1 - \delta$: $\text{err}_\mathcal{D}(\mathbb{Q}) \leq \text{err}_\mathcal{S}(\mathbb{Q}) + \sqrt{\frac{\text{KL}[\mathbb{Q}\|\mathbb{P}] + \log(2\sqrt{N}/\delta)}{2N}}$ where $\text{err}_\mathcal{D}(\mathbb{Q}) := \mathbb{E}_{(x,y) \sim \mathcal{D}, \theta \sim \mathbb{Q}}[\text{err}(f(x, \theta), y)]$, $\text{err}_\mathcal{S}(\mathbb{Q}) := \mathbb{E}_{(x,y) \sim \mathcal{S}, \theta \sim \mathbb{Q}}[\text{err}(f(x, \theta), y)]$, and $N$ denotes the cardinality of $\mathcal{S}$. That is, $\text{err}_\mathcal{D}(\mathbb{Q})$ and $\text{err}_\mathcal{S}(\mathbb{Q})$ are generalization error and empirical error, respectively. The only restriction on $\mathbb{P}$ here is that it cannot depend on the dataset $S$.

Following the recent discussion in Perez-Ortiz et al. (2021), one can construct data-dependent PAC-Bayes bounds by (i) randomly **partitioning** dataset $\mathcal{S}$ into $\mathcal{S}_\mathbb{Q}$ and $\mathcal{S}_\mathbb{P}$ so that they are independent, (ii) **pre-training** a PAC-Bayes prior distribution $\mathbb{P}_\mathcal{D}$ only dependent of $\mathcal{S}_\mathbb{P}$ (i.e., $\mathbb{P}_\mathcal{D}$ belongs to a PAC-Bayes prior due to the independence of $\mathcal{S}_\mathbb{Q}$), (iii) **fine-tuning** a PAC-Bayes posterior distribution $\mathbb{Q}$ dependent of entire dataset $\mathcal{S}$, and (iv) computing empirical error $\text{err}_{\mathcal{S}_\mathbb{Q}}(\mathbb{Q})$ with target subset $\mathcal{S}_\mathbb{Q}$ (not entire dataset $\mathcal{S}$). In summary, we modify the aforementioned original PAC-Bayes bound through a data-dependent prior $\mathbb{P}_\mathcal{D}$ as

$$\text{err}_\mathcal{D}(\mathbb{Q}) \leq \text{err}_{\mathcal{S}_\mathbb{Q}}(\mathbb{Q}) + \sqrt{\frac{\text{KL}[\mathbb{Q}\|\mathbb{P}_\mathcal{D}] + \log(2\sqrt{N_\mathbb{Q}}/\delta)}{2N_\mathbb{Q}}} \tag{2}$$

where $N_\mathbb{Q}$ is the cardinality of $\mathcal{S}_\mathbb{Q}$. We denote sets of input and output of partitioned datasets $(\mathcal{S}_\mathbb{P}, \mathcal{S}_\mathbb{Q})$ by $\mathcal{X}_\mathbb{P}, \mathcal{Y}_\mathbb{P}, \mathcal{X}_\mathbb{Q}, \mathcal{Y}_\mathbb{Q}$ for simplicity.

## 2.2 SCALE-INVARIANT PRIOR AND POSTERIOR FROM LINEARIZATION W.R.T. CONNECTIVITY

Our goal is to construct scale-invariant $\mathbb{P}_\mathcal{D}$ and $\mathbb{Q}$. To this end, we assume a pre-trained parameter $\theta^* \in \mathbb{R}^P$ with the prior dataset $\mathcal{S}_\mathbb{P}$. This parameter can be attained with standard NN optimization procedures (e.g., stochastic gradient descent (SGD) with momentum). Then, we consider a linearized NN at the pre-trained parameter with an auxiliary variable $c \in \mathbb{R}^P$ as

$$g_{\theta^*}^{\text{lin}}(x, c) := f(x, \theta^*) + \mathbf{J}_\theta(x, \theta^*)\text{diag}(\theta^*)c \tag{3}$$

where $\text{diag}$ is a vector-to-matrix diagonal operator. Note that equation 3 is the first-order Taylor approximation of NN with perturbation $\theta^* \odot c$ given input $x$ and parameter $\theta^*$: $g_{\theta^*}^{\text{pert}}(x, c) := f(x, \theta^* + \theta^* \odot c) \approx g_{\theta^*}^{\text{lin}}(x, c)$, where $\odot$ denotes element-wise multiplication of two vectors. Here we express the perturbation in parameter space as $\theta^* \odot c$ instead of a single variable such as $\delta \in \mathbb{R}^P$. By decomposing the scale and connectivity of perturbation, this linearization design matches the

scale of perturbation (i.e., $\theta^* \odot c$) to the scale of $\theta^*$ in a component-wise manner. Note that a similar decomposition was proposed in pruning at initialization (Lee et al., 2019c;b) to measure the importance of each connection independently of its weight. However, they only consider this form to predict the effect of each connection before pre-training.

Based on equation 3, we define a data-dependent prior ($\mathbb{P}_{\mathcal{D}}$) over connectivity as

$$\mathbb{P}_{\theta^*}(c) := \mathcal{N}(c \,|\, \mathbf{0}_P, \alpha^2 \mathbf{I}_P). \tag{4}$$

This distribution can be translated to a distribution over parameter by considering the distribution of perturbed parameter ($\psi := \theta^* + \theta^* \odot c$): $\mathbb{P}_{\theta^*}(\psi) := \mathcal{N}(\psi \,|\, \theta^*, \alpha^2 \mathrm{diag}(\theta^*)^2)$. We now define the PAC-Bayes posterior over connectivity $\mathbb{Q}(c)$ as follows:

$$\mathbb{Q}_{\theta^*}(c) := \mathcal{N}(c | \mu_{\mathbb{Q}}, \Sigma_{\mathbb{Q}}), \tag{5}$$

$$\mu_{\mathbb{Q}} := \frac{\Sigma_{\mathbb{Q}} \mathbf{J}_c^\top (\mathcal{Y} - f(\mathcal{X}, \theta^*))}{\sigma^2} = \frac{\Sigma_{\mathbb{Q}} \mathrm{diag}(\theta^*) \mathbf{J}_\theta^\top (\mathcal{Y} - f(\mathcal{X}, \theta^*))}{\sigma^2}, \tag{6}$$

$$\Sigma_{\mathbb{Q}} := \left( \frac{\mathbf{I}_P}{\alpha^2} + \frac{\mathbf{J}_c^\top \mathbf{J}_c}{\sigma^2} \right)^{-1} = \left( \frac{\mathbf{I}_P}{\alpha^2} + \frac{\mathrm{diag}(\theta^*) \mathbf{J}_\theta^\top \mathbf{J}_\theta \mathrm{diag}(\theta^*)}{\sigma^2} \right)^{-1} \tag{7}$$

where $\mathbf{J}_c \in NK \times P$ is a concatenation of $\mathbf{J}_c(x, \mathbf{0}_P) := \mathbf{J}_\theta(x, \theta^*) \mathrm{diag}(\theta^*) \in \mathbb{R}^{K \times P}$ (i.e., Jacobian of perturbed NN $g_{\theta^*}^{\mathrm{pert}}(x, c)$ w.r.t. $c$ at input $x$ and connectivity $\mathbf{0}_P$) along training input $\mathcal{X}$. Indeed, $\mathbb{Q}_{\theta^*}$ is *the posterior of Bayesian linear regression w.r.t. connectivity c*. We refer to Appendix D for detailed derivations. Again, it is equivalent to the posterior distribution over parameter $\mathbb{Q}_{\theta^*}(\psi) = \mathcal{N}\left(\psi | \theta^* + \theta^* \odot \mu_{\mathbb{Q}}, (\mathrm{diag}(\theta^*)^{-2}/\alpha^2 + \mathbf{J}_\theta^\top \mathbf{J}_\theta/\sigma^2)^{-1}\right)$ where $\mathrm{diag}(\theta^*)^{-2} := (\mathrm{diag}(\theta^*)^{-1})^2$ by assuming that all components of $\theta^*$ are non-zero. This assumption can be easily satisfied by considering the prior and posterior distributions of non-zero components of NNs only. Although we choose this restriction for theoretical tractability, future works can relax it to achieve diverse predictions by considering the distribution of zero coordinates.

In summary, a data-dependent PAC-Bayes bound can be computed with our PAC-Bayes distributions. The validity of this data-dependent PAC-Bayes bound is ensured as follows: our PAC-Bayes prior depends on the $\mathcal{S}_{\mathbb{P}}$ through $\theta^*$, but independent to the $\mathcal{S}_{\mathbb{Q}}$ that measures the errors. Note that here a two-phase training (i.e., **pre-training** with $\mathcal{S}_{\mathbb{P}}$ and **fine-tuning** with $\mathcal{S}$) explained in Sec. 2.1 is used to attain our PAC-Bayes posterior. Similar ideas of two-phase training with linearization were proposed in the context of transfer learning in Achille et al. (2021); Maddox et al. (2021). In transfer learning, there is a distribution shift between $\mathcal{S}_{\mathbb{P}}$ and $\mathcal{S}_{\mathbb{Q}}$. Therefore, $\mathcal{S}_{\mathbb{P}}$ cannot be used for their fine-tuning phase in contrast to our PAC-Bayes posterior.

Now we provide an invariance property of our prior and posterior distributions w.r.t. function-preserving scale transformations as follows: The main intuition behind this proposition is that **Jacobian w.r.t. connectivity is invariant to the function-preserving scaling transformation**, i.e., $\mathbf{J}_\theta(x, \mathcal{T}(\theta^*)) \mathrm{diag}(\mathcal{T}(\theta^*)) = \mathbf{J}_\theta(x, \theta^*) \mathrm{diag}(\theta^*)$. Representative cases of $\mathcal{T}$ in Proposition 2.1 are presented in Appendix E to highlight theoretical implications; these include the case of WD applied to the general network, including BN.

**Proposition 2.1** (Scale-invariance of PAC-Bayes prior and posterior). *Let $\mathcal{T} : \mathbb{R}^P \to \mathbb{R}^P$ is a invertible diagonal linear transformation such that $f(x, \mathcal{T}(\psi)) = f(x, \psi), \forall x \in \mathbb{R}^D, \forall \psi \in \mathbb{R}^P$. Then, both PAC-Bayes prior and posterior are invariant under $\mathcal{T}$:*

$$\mathbb{P}_{\mathcal{T}(\theta^*)}(c) \stackrel{d}{=} \mathbb{P}_{\theta^*}(c), \quad \mathbb{Q}_{\mathcal{T}(\theta^*)}(c) \stackrel{d}{=} \mathbb{Q}_{\theta^*}(c).$$

*Furthermore, generalization and empirical errors are also invariant to $\mathcal{T}$.*

## 2.3 RESULTING PAC-BAYES BOUND

Now we plug in our prior and posterior into the modified PAC-Bayes generalization error bound in equation 2. As a result, we obtain a novel generalization error bound, named **PAC-Bayes-CTK**, which is guaranteed to be invariant to scale transformations (hence without the contradiction of FM hypothesis mentioned in Sec. 1).

**Theorem 2.2** (PAC-Bayes-CTK and its invariance). *Let us assume pre-trained parameter $\theta^*$ with data $\mathcal{S}_{\mathbb{P}}$. By applying $\mathbb{P}_{\theta^*}$ and $\mathbb{Q}_{\theta^*}$ to data-dependent PAC-Bayes bound (equation 2), we get*

$$\mathrm{err}_{\mathcal{D}}(\mathbb{Q}_{\theta^*}) \leq \mathrm{err}_{\mathcal{S}_{\mathbb{Q}}}(\mathbb{Q}_{\theta^*}) + \sqrt{\overbrace{\underbrace{\frac{\mu_{\mathbb{Q}}^{\top}\mu_{\mathbb{Q}}}{4\alpha^2 N_{\mathbb{Q}}}}_{\text{(average) perturbation}} + \underbrace{\sum_{i=1}^{P}\frac{h(\beta_i)}{4N_{\mathbb{Q}}}}_{\text{sharpness}} + \frac{\log(2\sqrt{N_{\mathbb{Q}}}/\delta)}{2N_{\mathbb{Q}}}}^{\text{KL divergence}}} \tag{8}$$

*where $\{\beta_i\}_{i=1}^{P}$ are eigenvalues of $(\mathbf{I}_P + \frac{\alpha^2}{\sigma^2}\mathbf{J}_c^{\top}\mathbf{J}_c)^{-1}$ and $h(x) := x - \log(x) - 1$.* **This upper bound is invariant to $\mathcal{T}$ for the function-preserving scale transformation by Proposition 2.1.**

Note that recent works on FM contradiction focus only on the scale-invariance of sharpness metrics: **Indeed, their generalization bounds are not invariant to scale transformations due to the scale-dependent terms** (equation (34) in Tsuzuku et al. (2020) and equation (6) in Kwon et al. (2021)). Specifically, these terms are proportional to the norm of pre-trained parameters. On the other hand, the generalization bound in Petzka et al. (2021) (Theorem 11 in their paper) only holds for single-layer NNs, whereas our bound has no restrictions for network structure. As a result, our PAC-Bayes bound is the first scale-invariant PAC-Bayes bound to the best of our knowledge.

The following corollary explains why we name PAC-Bayes bound in Theorem 2.2 PAC-Bayes-CTK.

**Corollary 2.3** (Relation between CTK and PAC-Bayes-CTK). *Let us define empirical* **Connectivity Tangent Kernel** *(CTK) of $\mathcal{S}$ as $\mathbf{C}_{\mathcal{X}}^{\theta^*} := \mathbf{J}_c\mathbf{J}_c^{\top} = \mathbf{J}_{\theta}\mathrm{diag}(\theta^*)^2\mathbf{J}_{\theta}^{\top} \in \mathbb{R}^{NK \times NK}$ by removing below term. Note that empirical CTK has $T(\leq NK)$ non-zero eigenvalues of $\{\lambda_i\}_{i=1}^{T}$, then the followings hold for $\{\beta\}_{i=1}^{P}$ in Theorem 2.2: (i) $\beta_i = \sigma^2/(\sigma^2 + \alpha^2\lambda_i) < 1$ for $i = 1, \ldots, T$ and (ii) $\beta_i = 1$ for $i = T+1, \ldots, P$. Since $h(1) = 0$, this means $P - T$ terms of summation in the sharpness part of PAC-Bayes-CTK vanish to 0. Furthermore, this sharpness term of PAC-Bayes-CTK is a monotonically increasing function for each eigenvalue of empirical CTK.*

Corollary 2.3 clarifies why $\sum_{i=1}^{P} h(\beta_i)/4N_{\mathbb{Q}}$ in Theorem 2.2 is called the sharpness term of PAC-Bayes-CTK: A sharp pre-trained parameter would have large CTK eigenvalues (since eigenvalues of CTK measure the sensitivity of output w.r.t. connectivity), increasing the sharpness term and the generalization gap. Finally, Proposition 2.4 shows that empirical CTK is also scale-invariant.

**Proposition 2.4** (Scale-invariance of empirical CTK). *Let $\mathcal{T} : \mathbb{R}^P \to \mathbb{R}^P$ be an function-preserving scale transformation in Proposition 2.1. Then empirical CTK at parameter $\psi$ is invariant under $\mathcal{T}$:*

$$\mathbf{C}_{xy}^{\mathcal{T}(\psi)} := \mathbf{C}_{xy}^{\psi}, \,\forall x, y \in \mathbb{R}^D, \forall \psi \in \mathbb{R}^P. \tag{9}$$

**Remark 2.5** (Connections to empirical NTK). *The empirical CTK $\mathbf{C}_{xy}^{\psi}$ resembles the existing empirical Neural Tangent Kernel (NTK) at parameter $\psi$ (Jacot et al., 2018): $\Theta_{xy}^{\psi} := \mathbf{J}_{\theta}(x, \psi)\mathbf{J}_{\theta}(y, \psi)^{\top} \in \mathbb{R}^{K \times K}$. Note that the deterministic NTK in Jacot et al. (2018) is the infinite-width limiting kernel at initialized parameters, while empirical NTK can be defined on any (finite-width) NNs. We focus on empirical kernels for finite pre-trained parameters throughout the paper, and we leave deterministic kernels defined for future studies. Comparing empirical kernels, the main difference between empirical CTK and the existing empirical NTK is in the definition of Jacobian. In CTK, Jacobian is computed w.r.t. connectivity $c$ while the empirical NTK uses Jacobian w.r.t. parameters $\theta$. Therefore, another PAC-Bayes bound can be derived from the linearization of $f_{\theta^*}^{\mathrm{lin}}(x, \delta) := f(x, \theta^*) + \mathbf{J}_{\theta}(x, \theta^*)\delta$. As this bound is related to the eigenvalues of $\Theta_{\mathcal{X}}^{\theta^*}$, we call this bound* **PAC-Bayes-NTK** *and provide derivations in Appendix B. Note that PAC-Bayes-NTK is scale-variant as $\Theta_{xy}^{\mathcal{T}(\psi)} \neq \Theta_{xy}^{\psi}$ in general.*

## 2.4 Computing approximate bound in real world problems

To verify that PAC-Bayes bound in Theorem 2.2 is non-vacuous, we compute it for real-world problems. We use CIFAR-10 and 100 datasets (Krizhevsky, 2009), where the 50K training instances are randomly partitioned into $\mathcal{S}_{\mathbb{P}}$ of cardinality 45K and $\mathcal{S}_{\mathbb{Q}}$ of cardinality 5K. We refer to Appendix H for detailed experimental settings.

To compute equation 8, one needs (i) $\mu_{\mathbb{Q}}$-based perturbation term, (ii) $\mathbf{C}_{\mathcal{X}}^{\theta^*}$-based sharpness term, and (iii) samples from PAC-Bayes posterior $\mathbb{Q}_{\theta^*}$. $\mu_{\mathbb{Q}}$ in equation 6 can be obtained by minimizing

Table 1: Comparison between PAC-Bayes-CTK and PAC-Bayes-NTK for ResNet-18

| CIFAR-10 | PAC-Bayes-CTK | | | | PAC-Bayes-NTK | | | |
|---|---|---|---|---|---|---|---|---|
| Parameter scale | 0.5 | 1.0 | 2.0 | 4.0 | 0.5 | 1.0 | 2.0 | 4.0 |
| Trace ($\times 10^{-4}$) | 1.91 ± 0.04 | 1.91 ± 0.04 | 1.91 ± 0.04 | 1.91 ± 0.04 | 8793.18 ± 227.31 | 2590.97 ± 62.10 | 1107.58 ± 20.64 | 766.26 ± 11.80 |
| Perturbation | 6.26 ± 0.15 | 5.72 ± 0.09 | 5.77 ± 0.08 | 5.84 ± 0.05 | 636.55 ± 12.16 | 564.38 ± 7.56 | 438.21 ± 10.73 | 288.24 ± 6.38 |
| Sharpness | 28.92 ± 0.21 | 28.96 ± 0.22 | 28.95 ± 0.22 | 28.95 ± 0.20 | 728.80 ± 2.69 | 602.17 ± 2.77 | 502.32 ± 2.32 | 441.91 ± 2.08 |
| KL | 17.59 ± 0.17 | 17.34 ± 0.15 | 17.36 ± 0.15 | 17.39 ± 0.13 | 682.68 ± 4.74 | 583.27 ± 2.88 | 470.27 ± 4.32 | 365.07 ± 2.96 |
| Test err. ($\times 10^{2}$) | 4.82 ± 0.12 | 4.78 ± 0.11 | 4.78 ± 0.11 | 4.77 ± 0.12 | 13.00 ± 0.53 | 8.26 ± 0.17 | 6.94 ± 0.09 | 6.25 ± 0.08 |
| Bound ($\times 10^{2}$) | 9.21 ± 0.04 | 9.21 ± 0.02 | 9.21 ± 0.02 | 9.21 ± 0.03 | 39.00 ± 0.61 | 32.07 ± 0,25 | 28.24 ± 0.06 | 24.84 ± 0.11 |
| CIFAR-100 | PAC-Bayes-CTK | | | | PAC-Bayes-NTK | | | |
| Parameter scale | 0.5 | 1.0 | 2.0 | 4.0 | 0.5 | 1.0 | 2.0 | 4.0 |
| Trace ($\times 10^{-4}$) | 2.33 ± 0.37 | 2.34 ± 0.37 | 2.33 ± 0.37 | 2.34 ± 0.37 | 5830.55 ± 532.26 | 1913.90 ± 244.05 | 1089.53 ± 104.06 | 955.05 ± 66.16 |
| Perturbation | 14.54 ± 0.25 | 14.32 ± 0.24 | 14.08 ± 0.20 | 13.84 ± 0.14 | 620.18 ± 6.83 | 569.16 ± 6.94 | 459.16 ± 2.86 | 329.29 ± 3.34 |
| Sharpness | 42.52 ± 5.26 | 42.53 ± 5.26 | 42.52 ± 5.27 | 42.53 ± 5.26 | 694.45 ± 8.67 | 580.20 ± 12.22 | 519.78 ± 7.89 | 504.74 ± 6.10 |
| KL | 28.53 ± 2.51 | 28.42 ± 2.52 | 28.30 ± 2.55 | 28.19 ± 2.56 | 657.31 ± 7.21 | 574.68 ± 8.95 | 489.47 ± 3.57 | 417.02 ± 3.98 |
| Test err. ($\times 10^{2}$) | 21.78 ± 0.14 | 21.82 ± 00.18 | 21.84 ± 0.19 | 21.86 ± 0.21 | 43.39 ± 0.64 | 37.06 ± 0.26 | 32.32 ± 0.32 | 28.37 ± 0.13 |
| Bound ($\times 10^{2}$) | 27.74 ± 0.37 | 27.76 ± 0.40 | 27.75 ± 0.42 | 27.75 ± 0.42 | 68.44 ± 0.82 | 59.96 ± 0.19 | 51.90 ± 0.04 | 44.80 ± 0.25 |

$\arg\min_{c \in \mathbb{R}^P} L(c) = \frac{1}{2N} \|\mathcal{Y} - f(\mathcal{X}, \theta^*) - \mathbf{J}_c c\|^2 + \frac{\sigma^2}{2\alpha^2 N} c^\top c$ by first-order optimality condition. Note that this problem is a convex optimization problem w.r.t. $c$, since $c$ is the parameter of the linear regression problem. We use Adam optimizer (Kingma & Ba, 2014) with a fixed learning rate 1e-4 to solve this. For the sharpness term, we apply the Lanczos algorithm to approximate the eigenspectrum of $\mathbf{C}_{\mathcal{X}}^{\theta^*}$ following Ghorbani et al. (2019). We use 100 Lanczos iterations based on their setting. Lastly, we estimate empirical and test errors with 8 samples of CL/LL implementation of the Randomize-Then-Optimize (RTO) framework (Bardsley et al., 2014; Matthews et al., 2017). The pseudo-code and computational complexity of RTO implementation can be found in Appendix F.

Table 1 provides the bounds and related terms of PAC-Bayes-CTK (Theorem 2.2) and NTK (Theorem B.1). First, we found that our estimated PAC-Bayes-CTK and NTK are non-vacuous (i.e., estimated bounds are better than guessing at random) for ResNet-18 with 11M parameters. Note that deriving non-vacuous bound is challenging in PAC-Bayes analysis: only a few PAC-Bayes works (Dziugaite & Roy, 2017; Zhou et al., 2018; Perez-Ortiz et al., 2021) verified the non-vacuous property of their bounds, and other PAC-Bayes works (Foret et al., 2020; Tsuzuku et al., 2020) did not. To check the invariance property of PAC-Bayes-CTK, we scale the scale-invariant parameters in ResNet-18 (i.e., parameters preceding BN layers) for fixed constants $\{0.5, 1.0, 2.0, 4.0\}$. Due to BN layers, these transformations do not affect the function represented by NN, and the error bounds should be preserved for scale-invariant bounds. Table 1 shows that PAC-Bayes-CTK bound is stable to these transformations. On the other hand, PAC-Bayes-NTK bound is very sensitive to the parameter scale.

## 2.5 CONNECTIVITY SHARPNESS AND ITS EFFICIENT COMPUTATION

Now, we focus on the fact that the trace of CTK is also invariant to the parameter scale by Proposition 2.4. Unlike PAC-Bayes-CTK and NTK, traces of CTK and NTK do not require onerous hyper-parameter selection of $\delta, \alpha, \sigma$. Therefore, we simply define $\mathbf{CS}(\theta^*) := \text{tr}(\mathbf{C}_{\mathcal{X}}^{\theta^*})$ as a practical sharpness measure at $\theta^*$, named **Connectivity Sharpness** (CS) to detour the complex computation of PAC-Bayes-CTK. This metric can be easily applied to find NNs with better generalization, similar to other sharpness metrics (e.g., trace of Hessian), as shown in Jiang et al. (2020). We evaluate the detecting performance of CS in Sec. 4.1. The following corollary shows how CS can explain the generalization performance of NNs, conceptually.

**Corollary 2.6** (Connectivity sharpness, Informal). *Let us assume CTK and KL divergence terms of PAC-Bayes-CTK as defined in Theorem 2.2. Then, if CS vanishes to zero or infinity, the KL divergence term of PAC-Bayes-CTK also does so.*

As traces of a matrix can be efficiently estimated by Hutchinson's method (Hutchinson, 1989), one can compute the CS *without explicitly computing the entire CTK*. We refer to Appendix F for detailed procedures of computing CS. As CS is invariant to function-preserving scale transformations by Proposition 2.4, it does not contradict the FM hypothesis.

## 3 BAYESIAN NNS WITH SCALE-INVARIANCE

In this section, we discuss a practical implication of our posterior distribution (equation 5) used in the PAC-Bayes analysis. To this end, we first interpret our PAC-Bayes posterior as a modified result of

Table 2: Correlation analysis of sharpness measures with generalization gap. We refer Sec. 4.1 for the details of sharpness measures (row) and correlation metrics for sharpness-generalization relationship (column).

| | $\text{tr}(\mathbf{H})$ | $\text{tr}(\mathbf{F})$ | $\text{tr}(\mathbf{\Theta}^{\theta^*})$ | SO | PO | SM | PM | AS | FR | CS |
|---|---|---|---|---|---|---|---|---|---|---|
| $\tau$ (rank corr.) | 0.706 | 0.679 | 0.703 | 0.490 | 0.436 | 0.473 | 0.636 | 0.755 | 0.649 | **0.837** |
| network depth | 0.764 | 0.652 | **0.978** | -0.358 | -0.719 | 0.774 | 0.545 | 0.756 | 0.771 | **0.978** |
| network width | 0.687 | 0.922 | 0.330 | -0.533 | -0.575 | 0.495 | 0.564 | 0.827 | 0.921 | **0.978** |
| mini-batch size | 0.976 | 0.810 | **0.988** | 0.859 | 0.893 | 0.909 | 0.750 | 0.829 | 0.685 | 0.905 |
| learning rate | 0.966 | 0.713 | **1.000** | 0.829 | 0.874 | 0.057 | 0.621 | 0.794 | 0.565 | 0.897 |
| weight decay | -0.031 | -0.103 | 0.402 | 0.647 | 0.711 | 0.168 | 0.211 | 0.710 | 0.373 | **0.742** |
| $\Psi$ (avg.) | 0.672 | 0.599 | 0.739 | 0.289 | 0.237 | 0.481 | 0.538 | 0.783 | 0.663 | **0.900** |
| $\mathcal{K}$ (cond. MI) | 0.320 | 0.243 | 0.352 | 0.039 | 0.041 | 0.049 | 0.376 | 0.483 | 0.288 | **0.539** |

LA (MacKay, 1992). Then, we demonstrate that this modification improves existing LA when WD is applied to NNs with normalization layers (Proposition 3.1).

One can view the parameter space version of $\mathbb{Q}_{\theta^*}$ as a modified version of LA posterior (equation 1) by (i) substituting parameter-dependent damping $(\text{diag}(\theta^*)^{-2})$ for isotropic damping and (ii) adding perturbation $\theta^* \odot \mu_{\mathbb{Q}}$ to the mean of Gaussian distribution. Here, we focus on the effect of replacing the damping term of LA in batch-normalized NNs in the presence of WD. We refer to Antoran et al. (2021; 2022) for the discussion on the effect of adding perturbation to the LA with linearized NNs.

The main difference between the covariance terms of LA in equation 1 and equation 7 is the definition of Jacobian (i.e., parameter or connectivity) similar to the difference between empirical CTK and NTK in Remark 2.5. Therefore, we name $p_{\text{CL}}(\psi|\mathcal{S}) \sim \mathcal{N}(\psi|\theta^*, (\text{diag}(\theta^*)^{-2}/\alpha^2 + \mathbf{J}_\theta^\top \mathbf{J}_\theta/\sigma^2)^{-1})$ as **Connectivity Laplace (CL)** approximated posterior.

To compare CL posteriors against existing LAs, we explain how WD with BN can produce unexpected side effects of amplifying uncertainty. This side effect can be quantified if we consider linearized NN for LA, called Linearized Laplace (LL; Foong et al. (2019)). Assuming $\sigma^2 \ll \alpha^2$, the predictive distribution of LL and CL are

$$f_{\theta^*}^{\text{lin}}(x, \psi)|p_{\text{LA}}(\psi|\mathcal{S}) \sim \mathcal{N}(f(x, \theta^*), \alpha^2 \mathbf{\Theta}_{xx}^{\theta^*} - \alpha^2 \mathbf{\Theta}_{x\mathcal{X}}^{\theta^*} \mathbf{\Theta}_{\mathcal{X}}^{\theta^* -1} \mathbf{\Theta}_{\mathcal{X}x}^{\theta^*}) \tag{10}$$

$$f_{\theta^*}^{\text{lin}}(x, \psi)|p_{\text{CL}}(\psi|\mathcal{S}) \sim \mathcal{N}(f(x, \theta^*), \alpha^2 \mathbf{C}_{xx}^{\theta^*} - \alpha^2 \mathbf{C}_{x\mathcal{X}}^{\theta^*} \mathbf{C}_{\mathcal{X}}^{\theta^* -1} \mathbf{C}_{\mathcal{X}x}^{\theta^*}) \tag{11}$$

for any input $x \in \mathbb{R}^d$ where $\mathcal{X}$ in subscript means concatenation. We refer to Appendix G for the detailed derivations. The following proposition illustrates how WD with BN can increase the prediction uncertainty of equation 10.

**Proposition 3.1** (Uncertainty amplifying effect for LL). *Let us assume that $\mathcal{W}_\gamma : \mathbb{R}^P \to \mathbb{R}^P$ is a WD on scale-invariant parameters (e.g., parameters preceding BN layers) by multiplying $\gamma < 1$ and all the non-scale-invariant parameters are fixed. Then, the predictive uncertainty of LL is amplified by $1/\gamma^2 > 1$ while the predictive uncertainty of CTK is preserved as*

$$Var_{\psi \sim p_{\text{LA}}(\psi|\mathcal{S})}(f_{\mathcal{W}_\gamma(\theta^*)}^{\text{lin}}(x, \psi)) = Var_{\psi \sim p_{\text{LA}}(\psi|\mathcal{S})}(f_{\theta^*}^{\text{lin}}(x, \psi))/\gamma^2$$

$$Var_{\psi \sim p_{\text{CL}}(\psi|\mathcal{S})}(f_{\mathcal{W}_\gamma(\theta^*)}^{\text{lin}}(x, \psi)) = Var_{\psi \sim p_{\text{CL}}(\psi|\mathcal{S})}(f_{\theta^*}^{\text{lin}}(x, \psi))$$

*where $Var(\cdot)$ is variance of random variable.*

Since the primal regularization effect of WD actually occurs when combined with BN as experimentally shown in Zhang et al. (2019), Proposition 3.1 describes a real-world issue. Recently, Antoran et al. (2021; 2022) observed similar pitfalls in Proposition 3.1. However, their solution requires a more complicated hyper-parameter search: *independent prior selection for each normalized parameter group*. On the other hand, CL does not increase the hyper-parameter to be optimized compared to LL. We believe this difference will make CL more attractive to practitioners.

## 4 EXPERIMENTS

Here we describe experiments demonstrating (i) the effectiveness of Connectivity Sharpness (CS) as a generalization measurement metric and (ii) the usefulness of Connectivity Laplace (CL) as a general-purpose Bayesian NN: **With CS and CL, we can resolve the contradiction in the FM hypothesis concerning the generalization of NNs and attain stable calibration performance for various ranges of prior scales.**

Table 3: Test negative log-likelihood on two UCI variants (Hernández-Lobato & Adams, 2015; Foong et al., 2019). We marked the best method among the four in bold and marked the best method among LL/CL in italics.

| | Original (Hernández-Lobato & Adams, 2015) | | | | GAP variants (Foong et al., 2019) | | | |
|---|---|---|---|---|---|---|---|---|
| | Deep Ensemble | MCDO | LL | CL | Deep Ensemble | MCDO | LL | CL |
| boston_housing | 2.90 ± 0.03 | **2.63 ± 0.01** | 2.85 ± 0.01 | 2.88 ± 0.02 | 2.71 ± 0.01 | **2.68 ± 0.01** | 2.74 ± 0.01 | 2.75 ± 0.01 |
| concrete_strength | **3.06 ± 0.01** | 3.20 ± 0.00 | 3.22 ± 0.01 | 3.11 ± 0.02 | 4.03 ± 0.07 | **3.42 ± 0.00** | 3.47 ± 0.01 | 4.03 ± 0.02 |
| energy_efficiency | **0.74 ± 0.01** | 1.92 ± 0.01 | 2.12 ± 0.01 | 0.83 ± 0.01 | **0.77 ± 0.01** | 1.78 ± 0.01 | 2.02 ± 0.01 | 0.90 ± 0.02 |
| kin8nm | **-1.07 ± 0.00** | -0.80 ± 0.01 | -0.90 ± 0.00 | -1.07 ± 0.00 | **-0.94 ± 0.00** | -0.71 ± 0.00 | -0.87 ± 0.00 | -0.93 ± 0.00 |
| naval_propulsion | **-4.83 ± 0.00** | -3.85 ± 0.00 | -4.57 ± 0.00 | -4.76 ± 0.00 | -2.22 ± 0.33 | -3.36 ± 0.01 | -3.66 ± 0.11 | **-3.80 ± 0.07** |
| power_plant | **2.81 ± 0.00** | 2.91 ± 0.00 | 2.91 ± 0.00 | 2.81 ± 0.00 | 2.91 ± 0.00 | 2.97 ± 0.00 | 2.98 ± 0.00 | 2.91 ± 0.00 |
| protein_structure | **2.89 ± 0.00** | 2.96 ± 0.00 | 2.91 ± 0.00 | 2.89 ± 0.00 | 3.11 ± 0.00 | **3.07 ± 0.00** | 3.07 ± 0.00 | 3.13 ± 0.00 |
| wine | 1.21 ± 0.00 | **0.96 ± 0.01** | 1.24 ± 0.01 | 1.27 ± 0.01 | 1.48 ± 0.01 | **1.03 ± 0.00** | 1.45 ± 0.01 | 1.43 ± 0.00 |
| yacht_hydrodynamics | 1.26 ± 0.04 | 2.17 ± 0.06 | 1.20 ± 0.04 | 1.25 ± 0.04 | **1.71 ± 0.03** | 3.06 ± 0.02 | 1.78 ± 0.02 | 1.74 ± 0.01 |

## 4.1 CONNECTIVITY SHARPNESS AS A GENERALIZATION MEASUREMENT METRIC

Based on the CIFAR-10 dataset, we evaluate three correlation metrics to determine whether CS is more correlated with generalization performance than existing sharpness measures: (a) Kendall's rank-correlation coefficient ($\tau$; Kendall (1938)) (b) granulated Kendall's coefficient and their average ($\Psi$; Jiang et al. (2020)) (c) conditional independence test ($\mathcal{K}$; Jiang et al. (2020)). In all correlation metrics, a higher value indicates a stronger relationship between sharpness and generalization.

We compare CS to the following baseline sharpness measures: trace of Hessian ($\mathrm{tr}(\mathbf{H})$; Keskar et al. (2017)), trace of empirical Fisher ($\mathrm{tr}(\mathbf{F})$; Jastrzebski et al. (2021)), trace of empirical NTK at $\theta^*$, Fisher-Rao (FR; Liang et al. (2019)) metric, Adaptive Sharpness (AS; Kwon et al. (2021)), and four PAC-Bayes bound based measures: Sharpness-Orig. (SO), PAC-Bayes-Orig. (PO), Sharpness-Mag. (SM), and PAC-Bayes-Mag. (PM), which are eq. (52), (49), (62), (61) in Jiang et al. (2020). We compute granulated Kendall's correlation by using five hyper-parameters (network depth, network width, learning rate, weight decay, and mini-batch size) and three options for each. Thus, we train models with $3^5 = 243$ different training configurations. We vary the depth and width of NN based on VGG-13 (Simonyan & Zisserman, 2015). Further experimental details can be found in Appendix H.

In Table 2, CS shows the best results for $\tau$, $\Psi$, and $\mathcal{K}$ compared to all other sharpness measures. Additionally, granulated Kendall of CS is higher than other sharpness measures for 3 out of 5 hyperparameters and competitive with other sharpness measures for the remaining hyperparameters. The main difference between our CS and other sharpness measures is in the correlation with **weight decay/network width**: We found that SO and PM can capture the correlation with weight decay, and hypothesize that this is due to the weight norm term of SO/PO. As this weight norm term would interfere in capturing the sharpness-generalization correlation related to the number of parameters (i.e., width/depth), SO/PO fail to capture correlation with network width in Table 2. On the other hand, CS/AS do not suffer from such a problem. Also, it is notable that FR weakly captures this correlation despite its invariant property. For network width, we found that sharpness measures except for CS, $\mathrm{tr}(\mathbf{F})$, AS/FR fail to capture a strong correlation. **In summary, only CS/AS detect clear correlations with all hyperparameters; among them, CS captures clearer correlations.**

## 4.2 CONNECTIVITY LAPLACE AS AN EFFICIENT GENERAL-PURPOSE BAYESIAN NN

We evaluate CL's effectiveness as a general-purpose Bayesian NN using the UCI and CIFAR datasets. We refer to Appendix H for detailed experimental settings.

**UCI regression** We implement full-curvature versions of LL and CL and evaluate these to the 9 UCI regression datasets (Hernández-Lobato & Adams, 2015) and its GAP-variants (Foong et al., 2019) to compare calibration performance on *in-between* uncertainty. We measure test NLL for LL/CL and 2 baselines (Deep Ensemble (Lakshminarayanan et al., 2017) and Monte-Carlo DropOut (MCDO; Gal & Ghahramani (2016))). Eight ensemble members are used in Deep Ensemble, and 32 MC samples are used in LL, CL, and MCDO. Table 3 shows that CL performs better than LL on 6 of 9 datasets. Even though LL produces better calibration results on 3 of the datasets for both settings, the performance gaps between LL and CL are not as severe as on the other 6 datasets.

**Image classification.** We evaluate the uncertainty calibration performance of CL on CIFAR-10 and 100. As baseline methods, we consider Deterministic network, Monte-Carlo Dropout (MCDO; (Gal & Ghahramani, 2016)), Monte-Carlo Batch Normalization (MCBN; (Teye et al., 2018)), Deep Ensemble (Lakshminarayanan et al., 2017), Batch Ensemble (Wen et al., 2020), and LL (Khan et al.,

Table 4: Uncertainty calibration results on CIFAR-100 (Krizhevsky, 2009) for ResNet-18 (He et al., 2016)

| | CIFAR-100 | | | |
|---|---|---|---|---|
| | NLL (↓) | ECE (↓) | Brier. (↓) | AUC (↑) |
| Deterministic | 1.5370 ± 0.0117 | 0.1115 ± 0.0017 | 0.3889 ± 0.0031 | - |
| MCDO | 1.4264 ± 0.0110 | 0.0651 ± 0.0008 | 0.3925 ± 0.0020 | 0.6907 ± 0.0121 |
| MCBN | 1.4689 ± 0.0106 | 0.0998 ± 0.0016 | 0.3750 ± 0.0028 | 0.7982 ± 0.0210 |
| Batch Ensemble | 1.4029 ± 0.0031 | 0.0842 ± 0.0005 | 0.3582 ± 0.0010 | 0.7887 ± 0.0115 |
| Deep Ensemble | 1.0110 | 0.0507 | 0.2740 | 0.7802 |
| Linearized Laplace | 1.1673 ± 0.0093 | 0.0532 ± 0.0010 | 0.3597 ± 0.0020 | 0.8066 ± 0.0120 |
| Connectivity Laplace (Ours) | 1.1307 ± 0.0042 | 0.0524 ± 0.0019 | 0.3319 ± 0.0005 | 0.8423 ± 0.0204 |

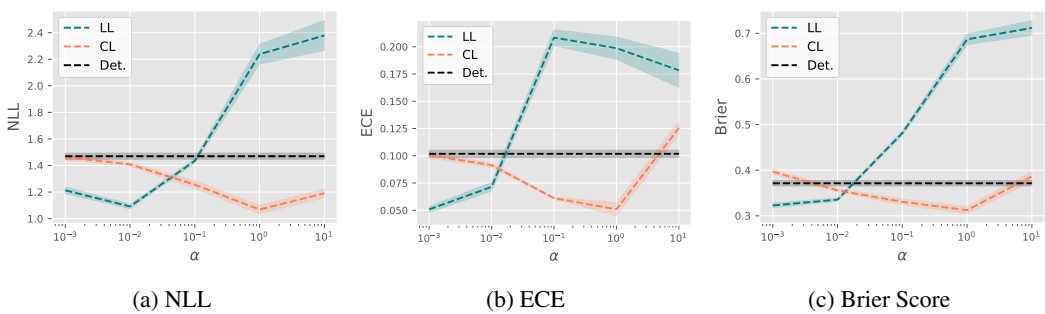

| (a) NLL | (b) ECE | (c) Brier Score |
|---|---|---|

Figure 1: Sensitivity to $\alpha$. Expected calibration error (ECE), Negative Log-likelihood (NLL), and Brier score results on corrupted CIFAR-100 for ResNet-18. Showing the mean (line) and standard deviation (shaded area) across four different seeds.

2019). We use the Randomize-Then-Optimize (RTO) implementation of LL/CL in Appendix F. We measure negative log-likelihood (NLL), expected calibration error (ECE; Guo et al. (2017)), and Brier score (Brier.) for ensemble predictions. We also measure the area under receiver operating curve (AUC) for OOD detection, where we set the SVHN (Netzer et al., 2011) dataset as an OOD.

Table 4 shows uncertainty calibration results on CIFAR-100. We refer to Appendix J for results on other settings, including CIFAR-10 and VGGNet. Except for Deep Ensemble, CL shows better results than baselines for all uncertainty calibration metrics. Deep Ensemble performs best in 3 out of 4 metrics, but each ensemble member requires full training. LL and CL, however, require only post-hoc training on pre-trained NNs. Particularly noteworthy is that CL presents competitive results with Deep Ensemble, even with much smaller computations.

**Robustness to the selection of prior scale.** Figure 1 shows the uncertainty calibration results over various $\alpha$ values for LL, CL, and Deterministic (Det.) baseline. As mentioned in previous works (Ritter et al., 2018; Kristiadi et al., 2020), the uncertainty calibration results of LL are extremely sensitive to the selection of $\alpha$. Especially, LL shows severe under-fitting for large $\alpha$ (i.e., small damping) regime. On the other hand, CL shows stable performance in the various ranges of $\alpha$.

## 5 CONCLUSION

In this work, we proposed a new approach to enhance the robustness of generalization bound using PAC-Bayes prior and posterior distributions. By separating scales and connectivities, our approach achieved invariance to function-preserving scale transformations, which is not addressed by existing generalization error bounds. As a result, our method successfully resolved the contradiction in the FM hypothesis caused by general scale transformation. In addition, our posterior distribution for PAC-Bayes analysis improved the Laplace approximation without significant drawbacks when dealing with weight decay with BN. To further improve our understanding of NN generalization effects, future research could explore extending prior and posterior distributions beyond Gaussian distributions to more task-specific distributions. This could help bridge the gap between theory and practice.

ACKNOWLEDGEMENTS

This work was supported by the Institute of Information & Communications Technology Planning & Evaluation (IITP) grant funded by the Korea government (MSIT) (No.2019-0-00075 / No.2017-0-01779) and the National Research Foundation of Korea (NRF) grants (No.2018R1A5A1059921) funded by the Korea government (MSIT). This work was also supported by Samsung Electronics Co., Ltd (No.IO201214-08133-01).

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

# A    NOTATIONS

Table 5: Notations used in the main paper

| | | |
|---:|:---:|:---|
| $x_n \in \mathbb{R}^D, y_n \in \mathbb{R}^K$ | $\triangleq$ | training inputs/outputs |
| $\theta, \psi \in \mathbb{R}^P$ | $\triangleq$ | parameter of NNs |
| $f(x, \theta)$ | $\triangleq$ | output of NN given $x, \theta$ |
| $f_{\theta^*}^{\text{lin}}(x, \delta)$ | $\triangleq$ | $f(x, \theta^*) + \mathbf{J}_\theta(x, \theta^*)\delta$ (linearization of NN w.r.t. parameter) |
| $g_{\theta^*}^{\text{lin}}(x, c)$ | $\triangleq$ | $f(x, \theta^*) + \mathbf{J}_\theta(x, \theta^*)\text{diag}(\theta^*)c$ (linearization of NN w.r.t. connectivity) |
| $f_{\theta^*}^{\text{pert}}(x, \delta), g_{\theta^*}^{\text{pert}}(x, c)$ | $\triangleq$ | Perturbed NN w.r.t. parameter/connectivity |
| $\mathbf{0}_d$ | $\triangleq$ | zero vector with dimension $d$ |
| $\mathbf{I}_d$ | $\triangleq$ | identity matrix with dimension $d \times d$ |
| $\text{tr}(A)$ | $\triangleq$ | a trace (i.e., sum of diagonal elements) of matrix $A$ |
| $\text{diag}(v)$ | $\triangleq$ | a diagonal matrix whose diagonal elements correspond to $v$ |
| $\mathcal{T} : \mathbb{R}^P \to \mathbb{R}^P$ | $\triangleq$ | a function-preserving scale transformation |
| $\mathcal{D}$ | $\triangleq$ | (true) data distribution |
| $\mathcal{S}$ | $\triangleq$ | i.i.d. sampled training dataset from $\mathcal{S}$ |
| $\mathcal{S}_\mathbb{P}, \mathcal{S}_\mathbb{Q}$ | $\triangleq$ | randomly partitioned prior/posterior datasets from $\mathcal{S}$ |
| $N, N_\mathbb{P}, N_\mathbb{Q}$ | $\triangleq$ | cardinality of $\mathcal{S}, \mathcal{S}_\mathbb{P}, \mathcal{S}_\mathbb{Q}$ |
| $\mathcal{X}, \mathcal{Y}/\mathcal{X}_\mathbb{P}, \mathcal{Y}_\mathbb{P}/\mathcal{X}_\mathbb{Q}, \mathcal{Y}_\mathbb{Q}$ | $\triangleq$ | concatenated inputs and outputs of all instances of $\mathcal{S}/\mathcal{S}_\mathbb{P}/\mathcal{S}_\mathbb{Q}$. |
| $\mathbf{J}_\theta(x, \theta^*), \mathbf{J}_c(x, \theta^*) \in \mathbb{R}^{K \times P}$ | $\triangleq$ | Jacobian of outputs w.r.t parameter/connectivity given $x, \theta^*$ |
| $f(\mathcal{X}, \theta) \in \mathbb{R}^{NK \times P}$ | $\triangleq$ | concatenated outputs of NN along $\mathcal{X}$ |
| $\mathbf{J}_\theta, \mathbf{J}_c \in \mathbb{R}^{NK \times P}$ | $\triangleq$ | concatenated Jacobian w.r.t. parameter/connectivity along $\mathcal{X}$ |
| $\mathbb{P}_{\theta^*}(c), \mathbb{Q}_{\theta^*}(c)$ | $\triangleq$ | our PAC-Bayes prior/posterior |
| $\text{err}_\mathcal{D}(\mathbb{Q}), \text{err}_\mathcal{S}(\mathbb{Q})$ | $\triangleq$ | generalization/empirical error of PAC-Bayes posterior |
| $\alpha$ | $\triangleq$ | standard deviation of our PAC-Bayes prior |
| $\sigma$ | $\triangleq$ | scale (standard deviation) of observational noise |
| $\mu_\mathbb{Q} \in \mathbb{R}^P, \Sigma_\mathbb{Q} \in \mathbb{R}^{P \times P}$ | $\triangleq$ | mean and covariance of our PAC-Bayes posterior |
| $\Theta_{xx'}^\psi, \mathbf{C}_{xx'}^\psi \in \mathbb{R}$ | $\triangleq$ | empirical NTK/CTK of NN given $\psi$ and input pair $x, x' \in \mathbb{R}^D$ |
| $\Theta_\mathcal{X}^\psi, \mathbf{C}_\mathcal{X}^\psi \in \mathbb{R}^{N \times N}$ | $\triangleq$ | empirical NTK/CTK of NN for training inputs ($\mathcal{X}$) given $\psi$ |
| $\{\lambda_i\}_{i=1}^P$ | $\triangleq$ | eigenvalues of empirical CTK |
| $\{\beta_i\}_{i=1}^P$ | $\triangleq$ | eigenvalues of $(\mathbf{I}_P + \frac{\alpha^2}{\sigma^2}\mathbf{J}_c^\top \mathbf{J}_c)^{-1}$ |
| $h(x)$ | $\triangleq$ | $x - \log(x) - 1$ (non-negative convex function w.r.t. $\beta_i$. See Fig. 2) |
| $(h \circ s)(x)$ | $\triangleq$ | non-negative concave function w.r.t. $\lambda_i$. See Fig. 2) |

# B    PROOFS

## B.1    PROOF OF PROPOSITION 2.1

*Proof.* Since the prior $\mathbb{P}_{\theta^*}(c)$ is independent to the parameter scale, $\mathbb{P}_{\theta^*}(c) \stackrel{d}{=} \mathbb{P}_{\mathcal{T}(\theta^*)}(c)$ is trivial. For Jacobian w.r.t. parameters, we have

$$\mathbf{J}_{\mathcal{T}(\theta)}(x, \mathcal{T}(\psi)) = \frac{\partial f(x, \mathcal{T}(\psi))}{\partial \mathcal{T}(\theta)} = \frac{\partial f(x, \psi)}{\partial \mathcal{T}(\theta)} = \frac{\partial f(x, \psi)}{\partial \theta}\frac{\partial \theta}{\partial \mathcal{T}(\theta)} = \mathbf{J}_\theta(x, \psi)\mathcal{T}^{-1}$$

Then, the Jacobian of NN w.r.t. connectivity at $\mathcal{T}(\psi)$ holds

$$\mathbf{J}_\theta(x, \mathcal{T}(\psi))\text{diag}(\mathcal{T}(\psi)) = \mathbf{J}_\theta(x, \psi)\mathcal{T}^{-1}\mathcal{T}\text{diag}(\psi) \tag{12}$$
$$= \mathbf{J}_\theta(x, \psi)\text{diag}(\psi) \tag{13}$$

where the first equality holds from the above one and the fact that $\mathcal{T}$ is a diagonal linear transformation. Therefore, the covariance of posterior is invariant to $\mathcal{T}$.

$$
\left( \frac{\mathbf{I}_P}{\alpha^2} + \frac{\mathrm{diag}(\mathcal{T}(\theta^*))\mathbf{J}_\theta^\top(\mathcal{X}, \mathcal{T}(\theta^*))\mathbf{J}_\theta(\mathcal{X}, \mathcal{T}(\theta^*))\mathrm{diag}(\mathcal{T}(\theta^*))}{\sigma^2} \right)^{-1}
$$

$$
= \left( \frac{\mathbf{I}_P}{\alpha^2} + \frac{\mathrm{diag}(\theta^*)\mathbf{J}_\theta^\top(\mathcal{X}, \theta^*)\mathbf{J}_\theta(\mathcal{X}, \theta^*)\mathrm{diag}(\theta^*)}{\sigma^2} \right)^{-1}
$$

$$
= \left( \frac{\mathbf{I}_P}{\alpha^2} + \frac{\mathrm{diag}(\theta^*)\mathbf{J}_\theta^\top \mathbf{J}_\theta \mathrm{diag}(\theta^*)}{\sigma^2} \right)^{-1}
$$

Moreover, the mean of posterior is also invariant to $\mathcal{T}$.

$$
\frac{\Sigma_\mathbb{Q}\mathrm{diag}(\mathcal{T}(\theta^*))\mathbf{J}_\theta^\top(\mathcal{X}, \mathcal{T}(\theta^*))\left(\mathcal{Y} - f(\mathcal{X}, \mathcal{T}(\theta^*))\right)}{\sigma^2}
$$

$$
= \frac{\Sigma_\mathbb{Q}\mathrm{diag}(\mathcal{T}(\theta^*))\mathbf{J}_\theta^\top(\mathcal{X}, \mathcal{T}(\theta^*))\left(\mathcal{Y} - f(\mathcal{X}, \theta^*)\right)}{\sigma^2}
$$

$$
= \frac{\Sigma_\mathbb{Q}\mathrm{diag}(\theta^*)\mathbf{J}_\theta^\top(\mathcal{X}, \theta^*)\left(\mathcal{Y} - f(\mathcal{X}, \theta^*)\right)}{\sigma^2}
$$

$$
= \frac{\Sigma_\mathbb{Q}\mathrm{diag}(\theta^*)\mathbf{J}_\theta^\top\left(\mathcal{Y} - f(\mathcal{X}, \theta^*)\right)}{\sigma^2}
$$

Therefore, equation 6 and equation 7 are invariant to function-preserving scale transformations. The remaining part of the proposition is related to the definition of function-preserving scale transformation $\mathcal{T}$. For generalization error, the following holds

$$
\begin{aligned}
\mathrm{err}_\mathcal{D}(\mathbb{Q}_{\mathcal{T}(\theta^*)}) &= \mathbb{E}_{(x,y)\sim\mathcal{D}, \psi\sim\mathbb{Q}_{\mathcal{T}(\theta^*)}}[\mathrm{err}(f(x,\psi), y)] \\
&= \mathbb{E}_{(x,y)\sim\mathcal{D}, c\sim\mathbb{Q}_{\mathcal{T}(\theta^*)}}[\mathrm{err}(g_{\theta^*}^{\mathrm{pert}}(x,c), y)] \\
&= \mathbb{E}_{(x,y)\sim\mathcal{D}, c\sim\mathbb{Q}_{\theta^*}}[\mathrm{err}(g_{\theta^*}^{\mathrm{pert}}(x,c), y)] \\
&= \mathbb{E}_{(x,y)\sim\mathcal{D}, \psi\sim\mathbb{Q}_{\theta^*}}[\mathrm{err}(f(x,\psi), y)] \\
&= \mathrm{err}_\mathcal{D}(\mathbb{Q}_{\theta^*})
\end{aligned}
$$

WLOG, this proof can be extended to the empirical error $\mathrm{err}_{\mathcal{S}_\mathbb{Q}}$. $\qquad\square$

## B.2 Proof of Theorem 2.2

*Proof.* **(Construction of KL divergence)** To construct PAC-Bayes-CTK, we need to arrange KL divergence between posterior and prior as follows:

$$
\begin{aligned}
\mathrm{KL}[\mathbb{Q}\|\mathbb{P}] &= \frac{1}{2}\left(\mathrm{tr}\left(\Sigma_\mathbb{P}^{-1}(\Sigma_\mathbb{Q} + (\mu_\mathbb{Q} - \mu_\mathbb{P})(\mu_\mathbb{Q} - \mu_\mathbb{P})^\top)\right) + \log|\Sigma_\mathbb{P}| - \log|\Sigma_\mathbb{Q}| - P\right) \\
&= \frac{1}{2}\mathrm{tr}(\Sigma_\mathbb{P}^{-1}(\mu_\mathbb{Q} - \mu_\mathbb{P})(\mu_\mathbb{Q} - \mu_\mathbb{P})^\top)) + \frac{1}{2}\left(\mathrm{tr}(\Sigma_\mathbb{P}^{-1}\Sigma_\mathbb{Q}) + \log|\Sigma_\mathbb{P}| - \log|\Sigma_\mathbb{Q}| - P\right) \\
&= \frac{1}{2}(\mu_\mathbb{Q} - \mu_\mathbb{P})^\top\Sigma_\mathbb{P}^{-1}(\mu_\mathbb{Q} - \mu_\mathbb{P}) + \frac{1}{2}\left(\mathrm{tr}(\Sigma_\mathbb{P}^{-1}\Sigma_\mathbb{Q}) - \log|\Sigma_\mathbb{P}^{-1}\Sigma_\mathbb{Q}| - P\right) \\
&= \underbrace{\frac{\mu_\mathbb{Q}^\top\mu_\mathbb{Q}}{2\alpha^2}}_{\text{perturbation}} + \underbrace{\frac{1}{2}\left(\mathrm{tr}(\Sigma_\mathbb{P}^{-1}\Sigma_\mathbb{Q}) - \log|\Sigma_\mathbb{P}^{-1}\Sigma_\mathbb{Q}| - p\right)}_{\text{sharpness}}
\end{aligned}
$$

where the first equality uses the KL divergence between two Gaussian distributions, the third equality uses trace property ($\mathrm{tr}(AB) = \mathrm{tr}(BA)$ and $\mathrm{tr}(a) = a$ for scalar $a$), and the last equality uses the definition of PAC-Bayes prior ($\mathbb{P}_{\theta^*}(c) = \mathcal{N}(c|\mathbf{0}_P, \alpha^2\mathbf{I}_P)$). For sharpness term, we first compute the $\Sigma_\mathbb{P}^{-1}\Sigma_\mathbb{Q}$ term as

$$
\Sigma_\mathbb{P}^{-1}\Sigma_\mathbb{Q} = \left(\mathbf{I}_P + \frac{\alpha^2}{\sigma^2}\mathbf{J}_c^\top\mathbf{J}_c\right)^{-1}
$$

Since $\alpha^2, \sigma^2 > 0$ and $\mathbf{J}_c^\top \mathbf{J}_c$ is positive semi-definite, the matrix $\Sigma_\mathbb{P}^{-1}\Sigma_\mathbb{Q}$ have non-zero eigenvalues of $\{\beta_i\}_{i=1}^P$. Since a trace is the sum of eigenvalues and the log-determinant is the sum of the log of eigenvalues, we have

$$\mathrm{KL}[\mathbb{Q}\|\mathbb{P}] = \frac{\mu_\mathbb{Q}^\top \mu_\mathbb{Q}}{2\alpha^2} + \frac{1}{2}\sum_{i=1}^P (\beta_i - \log(\beta_i) - 1) = \frac{\mu_\mathbb{Q}^\top \mu_\mathbb{Q}}{2\alpha^2} + \frac{1}{2}\sum_{i=1}^P h(\beta_i)$$

where $h(x) = x - \log(x) - 1$. By plugging this KL divergence into the equation 2, we get equation 8.

(**Eigenvalues of $\Sigma_\mathbb{P}^{-1}\Sigma_\mathbb{Q}$**) To show the scale-invariance of PAC-Bayes-CTK, it is sufficient to show that the KL divergence between posterior and prior is scale-invariant: $\log(2\sqrt{N_\mathbb{Q}}/\delta)/2N_\mathbb{Q}$ is independent to KL PAC-Bayes prior/posterior. We already show the invariance property of empirical/generalization error term in Proposition 2.1.

To show the invariance property of KL divergence, let us write a singular value decomposition of Jacobian w.r.t. connectivity $\mathbf{J}_c \in \mathbb{R}^{NK \times P}$ as $\mathbf{J}_c = U\Sigma V^\top$, where $U \in \mathbb{R}^{NK \times NK}$ and $V \in \mathbb{R}^{P \times P}$ are orthogonal matrices and $\Sigma \in \mathbb{R}^{NK \times P}$ is a rectangular diagonal matrix with descending order for singular values. Then, the following holds for $\Sigma_\mathbb{P}^{-1}\Sigma_\mathbb{Q}$

$$\Sigma_\mathbb{P}^{-1}\Sigma_\mathbb{Q} = \left(\mathbf{I}_P + \frac{\alpha^2}{\sigma^2}\mathbf{J}_c^\top \mathbf{J}_c\right)^{-1}$$

$$= \left(\mathbf{I}_P + \frac{\alpha^2}{\sigma^2}V\Sigma^\top \Sigma V^\top\right)^{-1}$$

$$= V\left(\mathbf{I}_P + \frac{\alpha^2}{\sigma^2}\Lambda\right)^{-1}V^\top$$

where $\Lambda = \Sigma^\top \Sigma \in \mathbb{R}^{P \times P}$ is a diagonal matrix with $\lambda_i := \Lambda_{ii} = 0$ for $i \geq NK$. Therefore, eigenvalues of $\Sigma_\mathbb{P}^{-1}\Sigma_\mathbb{Q}$ are $\frac{1}{1+\alpha^2\lambda_i/\sigma^2} = \frac{\sigma^2}{\sigma^2+\alpha^2\lambda_i}$. Now, we consider Connectivity Tangent Kernel (CTK) as defined in equation 2.3:

$$\mathbf{C}_\mathcal{X}^{\theta^*} := \mathbf{J}_c\mathbf{J}_c^\top = \mathbf{J}_\theta \mathrm{diag}(\theta^*)^2\mathbf{J}_\theta^\top \in \mathbb{R}^{NK \times NK}.$$

Similar to $\mathbf{J}_c^\top \mathbf{J}_c$, CTK can be expressed as follows

$$\mathbf{C}_\mathcal{X}^{\theta^*} = \mathbf{J}_c\mathbf{J}_c^\top = U\Sigma V^\top V\Sigma^\top U^\top = U\Sigma\Sigma^\top U^\top = U\Lambda' U^\top$$

where $\Lambda' = \Sigma\Sigma^\top \in \mathbb{R}^{NK \times NK}$. As the smallest $(P - NK)$ eigenvalues of $\Lambda = \Sigma^\top \Sigma$ are just zeros, $\Lambda'$ is just a reduced diagonal matrix of $\Lambda$ with these eigenvalues removed. As a result, $\{\lambda_i\}_{i=1}^{NK}$ are eigenvalues of CTK.

(**Scale invariance of CTK**) The scale-invariance property of CTK is a simple application of equation 13:

$$\mathbf{C}_{xy}^{\mathcal{T}(\psi)} = \mathbf{J}_{\mathcal{T}(\theta)}(x, \mathcal{T}(\psi))\mathrm{diag}(\mathcal{T}(\psi)^2)\mathbf{J}_{\mathcal{T}(\theta)}(y, \mathcal{T}(\psi))^\top$$

$$= \mathbf{J}_\theta(x, \psi)\mathcal{T}^{-1}\mathcal{T}\mathrm{diag}(\psi)\mathrm{diag}(\psi)\mathcal{T}\mathcal{T}^{-1}\mathbf{J}_\theta(x, \psi)^\top$$

$$= \mathbf{J}_\theta(x, \psi)\mathrm{diag}(\psi)\mathrm{diag}(\psi)\mathbf{J}_\theta(x, \psi)^\top$$

$$= \mathbf{C}_{xy}^\psi, \forall x, y \in \mathbb{R}^D, \forall \psi \in \mathbb{R}^P.$$

Therefore, CTK is invariant to any function-preserving scale transformation $\mathcal{T}$ and so do its eigenvalues. This guarantees the invariance of $\Sigma_\mathbb{P}^{-1}\Sigma_\mathbb{Q}$ and its eigenvalues. In summary, we showed the scale-invariance property of the sharpness term of KL divergence. Now all that remains is to show the invariance of the perturbation term. However, this is already proved in the proof of Proposition 2.1. Therefore, we show PAC-Bayes-CTK is invariant to any function-preserving scale transformation. $\square$

### B.3 Proof of Corollary 2.3

*Proof.* In proof of Theorem 2.2, we showed that eigenvalues of $\Sigma_\mathbb{P}^{-1}\Sigma_\mathbb{Q}$ can be represented as

$$\frac{\sigma^2}{\sigma^2 + \alpha^2\lambda_i}$$

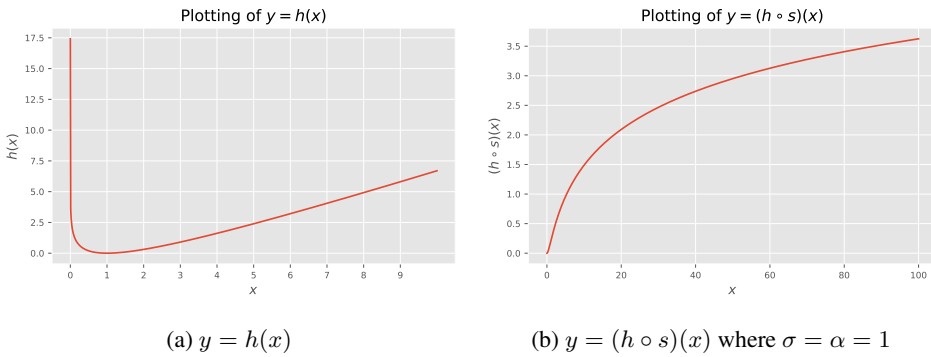

(a) $y = h(x)$          (b) $y = (h \circ s)(x)$ where $\sigma = \alpha = 1$

Figure 2: Functions used in proofs

where $\{\lambda_i\}_{i=1}^{P}$ are eigenvalues of $\Lambda = \Sigma^\top \Sigma$. By SVD of Jacobian w.r.t. connectivity, $\lambda_i = \Lambda_{ii} = \Sigma_{ii}^2 \geq 0$. Therefore, eigenvalues of CTK are squares of singular values of $\mathbf{J}_c$ and so $\lambda_i \geq 0, \forall i$. As a result, $\beta_i \leq 1$ for all $i = 1, \ldots, P$ for eigenvalues $\{\beta_i\}_{i=1}^{P}$ of $\Sigma_{\mathbb{P}}^{-1}\Sigma_{\mathbb{Q}}$ and equality holds for $\lambda_i = 0$ (i.e., $i \geq NK$). Now all that remains is to show that the sharpness term of PAC-Bayes-CTK is a monotonically increasing function on each eigenvalue of CTK. To show this, we first keep in mind that

$$ s(x) := \frac{\sigma^2}{\sigma^2 + \alpha^2 x} $$

is a monotonically decreasing function for $x \geq 0$ and $h(x) := x - \log(x) - 1$ is a monotonically decreasing function for $x \in (0, 1]$. Since the sharpness term of KL divergence is

$$ \sum_{i=1}^{P} \frac{h(\beta_i)}{4N_{\mathbb{Q}}} = \sum_{i=1}^{P} \frac{(h \circ s)(\lambda_i)}{4N_{\mathbb{Q}}} $$

This is a monotonically increasing function for $x \geq 0$ since $s(x) \leq 1$ for $x \geq 0$. For your information, we plot $y = h(x)$ and $y = (h \circ s)(x)$ in Figure 2.

$\square$

### B.4   PROOF OF PROPOSITION 2.4

We refer to **Scale invariance of CTK** part of the proof of Theorem 2.2. This is a direct application of the scale-invariance property of Jacobian w.r.t. connectivity.

### B.5   PROOF OF COROLLARY 2.6

*Proof.* Since CS is a trace of CTK, it is a sum of the eigenvalues of CTK. As shown in the proof of Corollary 2.3, eigenvalues of CTK are square of singular values of Jacobian w.r.t. connectivity $c$. Therefore, the eigenvalues of CTK are non-negative and vanish to zero if CS vanishes to zero.

$$ \sum_{i=1}^{P} \lambda_i = 0 \Rightarrow \lambda_i = 0 \Rightarrow \beta_i = s(\lambda_i) = 1 \Rightarrow h(\beta_i) = 0, \quad \forall i = 1, \ldots, P $$

This means the sharpness term of KL divergence vanishes to zero. Furthermore, singular values of Jacobian w.r.t. $c$ also vanish to zero in this case. Therefore, $\mu_{\mathbb{Q}}$ vanishes to zero, also. Similarly, if CS diverges to infinity, this means (at least) one of the eigenvalues of CTK diverges to infinity. In this case, the following holds

$$ \lambda_i \to \infty \Rightarrow \beta_i = s(\lambda_i) \to 0 \Rightarrow h(\beta_i) \to \infty, \quad \forall i = 1, \ldots, P $$

Therefore, the KL divergence term of PAC-Bayes-CTK also diverges to infinity. $\square$

## B.6 PROOF OF PROPOSITION 3.1

*Proof.* By assumption, we fixed all non-scale invariant parameters. This means we exclude these parameters in the sampling procedure of CL and LL. In terms of predictive distribution, this can be translated as

$$f_{\theta^*}^{\text{lin}}(x,\psi)|p_{\text{LA}}(\psi|\mathcal{S}) \sim \mathcal{N}(f(x,\theta^*), \alpha^2 \hat{\Theta}_{xx}^{\theta^*} - \alpha^2 \hat{\Theta}_{x\mathcal{X}}^{\theta^*} \hat{\Theta}_{\mathcal{X}}^{\theta^* -1} \hat{\Theta}_{\mathcal{X}x}^{\theta^*})$$
$$f_{\theta^*}^{\text{lin}}(x,\psi)|p_{\text{CL}}(\psi|\mathcal{S}) \sim \mathcal{N}(f(x,\theta^*), \alpha^2 \hat{\mathbf{C}}_{xx}^{\theta^*} - \alpha^2 \hat{\mathbf{C}}_{x\mathcal{X}}^{\theta^*} \hat{\mathbf{C}}_{\mathcal{X}}^{\theta^* -1} \hat{\mathbf{C}}_{\mathcal{X}x}^{\theta^*})$$

where $\hat{\Theta}_{xx'}^{\psi} := \sum_{i \in \mathcal{P}} \frac{\partial f(x,\psi)}{\partial \theta_i} \frac{\partial f(x',\psi)}{\partial \theta_i}$ and $\hat{\mathbf{C}}_{xx'}^{\psi} := \sum_{i \in \mathcal{P}} \frac{\partial f(x,\psi)}{\partial \theta_i} \frac{\partial f(x',\psi)}{\partial \theta_i}(\psi_i)^2$ for scale-invariant parameter set $\mathcal{P}$. Thereby, we mask the gradient of the non-scale-invariant parameters as zero. Therefore, this can be arranged as follows

$$\hat{\Theta}_{xx'}^{\psi} = \mathbf{J}_\theta(x,\psi)\text{diag}(\mathbf{1}_\mathcal{P})\mathbf{J}_\theta(x,\psi)^\top, \quad \hat{\mathbf{C}}_{xx'}^{\psi} = \mathbf{J}_\theta(x,\psi)\text{diag}(\psi)\text{diag}(\mathbf{1}_\mathcal{P})\text{diag}(\psi)\mathbf{J}_\theta(x,\psi)^\top$$

where $\mathbf{1}_\mathcal{P} \in \mathbb{R}^P$ is a masking vector (i.e., one for included components and zero for excluded components). Then, the WD for scale-invariant parameters can be represented as

$$\mathcal{W}_\gamma(\psi)_i = \begin{cases} \gamma\psi_i, & \text{if} \quad \psi_i \in \mathcal{P}. \\ \psi_i, & \text{if} \quad \psi_i \notin \mathcal{P}. \end{cases}$$

Therefore, we get

$$\hat{\Theta}_{xx'}^{\mathcal{W}_\gamma(\psi)} = \mathbf{J}_\theta(x,\mathcal{W}_\gamma(\psi))\text{diag}(\mathbf{1}_\mathcal{P})\mathbf{J}_\theta(x,\mathcal{W}_\gamma(\psi)))^\top$$
$$= \mathbf{J}_\theta(x,\psi)\mathcal{W}_\gamma^{-1}\text{diag}(\mathbf{1}_\mathcal{P})\mathcal{W}_\gamma^{-1}\mathbf{J}_\theta(x,\psi)^\top$$
$$= \mathbf{J}_\theta(x,\psi)\text{diag}(\mathbf{1}_\mathcal{P}/\gamma^2)\mathbf{J}_\theta(x,\psi)^\top$$
$$= 1/\gamma^2\mathbf{J}_\theta(x,\psi)\text{diag}(\mathbf{1}_\mathcal{P})\mathbf{J}_\theta(x,\psi)^\top$$
$$= 1/\gamma^2\hat{\Theta}_{xx'}^{\psi}$$

for empirical NTK and

$$\hat{\mathbf{C}}_{xx'}^{\mathcal{W}_\gamma(\psi)} = \mathbf{J}_\theta(x,\mathcal{W}_\gamma(\psi))\text{diag}(\mathcal{W}_\gamma(\psi))\text{diag}(\mathbf{1}_\mathcal{P})\text{diag}(\mathcal{W}_\gamma(\psi))\mathbf{J}_\theta(x,\mathcal{W}_\gamma(\psi)))^\top$$
$$= \mathbf{J}_\theta(x,\psi)\mathcal{W}_\gamma^{-1}\mathcal{W}_\gamma\text{diag}(\psi)\text{diag}(\mathbf{1}_\mathcal{P})\text{diag}(\psi)\mathcal{W}_\gamma\mathcal{W}_\gamma^{-1}\mathbf{J}_\theta(x,\psi)^\top$$
$$= \mathbf{J}_\theta(x,\psi)\text{diag}(\psi)\text{diag}(\mathbf{1}_\mathcal{P})\text{diag}(\psi)\mathbf{J}_\theta(x,\psi)^\top$$
$$= \hat{\mathbf{C}}_{xx'}^{\psi}$$

for empirical CTK. Therefore, we get

$$f_{\mathcal{W}_\gamma(\theta^*)}^{\text{lin}}(x,\psi)|p_{\text{LA}}(\psi|\mathcal{S}) \sim \mathcal{N}(f(x,\theta^*), \alpha^2/\gamma^2 \hat{\Theta}_{xx}^{\theta^*} - \alpha^2/\gamma^2 \hat{\Theta}_{x\mathcal{X}}^{\theta^*} \hat{\Theta}_{\mathcal{X}}^{\theta^* -1} \hat{\Theta}_{\mathcal{X}x}^{\theta^*})$$
$$f_{\mathcal{W}_\gamma(\theta^*)}^{\text{lin}}(x,\psi)|p_{\text{CL}}(\psi|\mathcal{S}) \sim \mathcal{N}(f(x,\theta^*), \alpha^2 \hat{\mathbf{C}}_{xx}^{\theta^*} - \alpha^2 \hat{\mathbf{C}}_{x\mathcal{X}}^{\theta^*} \hat{\mathbf{C}}_{\mathcal{X}}^{\theta^* -1} \hat{\mathbf{C}}_{\mathcal{X}x}^{\theta^*})$$

This gives us

$$\text{Var}_{\psi \sim p_{\text{LA}}(\psi|\mathcal{S})}(f_{\mathcal{W}_\gamma(\theta^*)}^{\text{lin}}(x,\psi)) = \text{Var}_{\psi \sim p_{\text{LA}}(\psi|\mathcal{S})}(f_{\theta^*}^{\text{lin}}(x,\psi))/\gamma^2$$
$$\text{Var}_{\psi \sim p_{\text{CL}}(\psi|\mathcal{S})}(f_{\mathcal{W}_\gamma(\theta^*)}^{\text{lin}}(x,\psi)) = \text{Var}_{\psi \sim p_{\text{CL}}(\psi|\mathcal{S})}(f_{\theta^*}^{\text{lin}}(x,\psi))$$

$\square$

## B.7 DERIVATION OF PAC-BAYES-NTK

**Theorem B.1** (PAC-Bayes-NTK). *Let us assume pre-trained parameter $\theta^*$ with data $\mathcal{S}_\mathbb{P}$. Let us assume PAC-Bayes prior and posterior as*

$$\mathbb{P}'_{\theta*}(\delta) := \mathcal{N}(\delta|\mathbf{0}_P, \alpha^2\mathbf{I}_P) \tag{14}$$

$$\mathbb{Q}'_{\theta*}(\delta) := \mathcal{N}(\delta|\mu_{\mathbb{Q}'}, \Sigma_{\mathbb{Q}'}) \tag{15}$$

$$\mu_{\mathbb{Q}'} := \frac{\Sigma_{\mathbb{Q}'}\mathbf{J}_\theta^\top(\mathcal{Y} - f(\mathcal{X},\theta^*))}{\sigma^2} \tag{16}$$

$$\Sigma_{\mathbb{Q}'} := \left(\frac{\mathbf{I}_P}{\alpha^2} + \frac{\mathbf{J}_\theta^\top\mathbf{J}_\theta}{\sigma^2}\right)^{-1} \tag{17}$$

*By applying $\mathbb{P}'_{\theta^*}, \mathbb{Q}'_{\theta^*}$ to data-dependent PAC-Bayes bound (equation 2), we get*

$$\mathrm{err}_{\mathcal{D}}(\mathbb{Q}'_{\theta^*}) \leq \mathrm{err}_{\mathcal{S}_{\mathbb{Q}'}}(\mathbb{Q}'_{\theta^*}) + \sqrt{\overbrace{\underbrace{\frac{\mu_{\mathbb{Q}'}^\top \mu_{\mathbb{Q}'}}{4\alpha^2 N_{\mathbb{Q}'}}}_{\text{(average) perturbation}} + \underbrace{\sum_{i=1}^{P} \frac{h\left(\beta'_i\right)}{4N_{\mathbb{Q}'}}}_{\text{sharpness}}}^{\text{KL divergence}} + \frac{\log(2\sqrt{N_{\mathbb{Q}'}}/\delta)}{2N_{\mathbb{Q}'}}} \qquad (18)$$

*where $\{\beta'_i\}_{i=1}^{P}$ are eigenvalues of $(\mathbf{I}_P + \frac{\alpha^2}{\sigma^2}\mathbf{J}_\theta^\top \mathbf{J}_\theta)^{-1}$ and $h(x) := x - \log(x) - 1$. This upper bound is not scale-invariant in general.*

*Proof.* The main difference between PAC-Bayes-CTK and PAC-Bayes-NTK is the definition of Jacobian: PAC-Bayes-CTK uses Jacobian w.r.t connectivity and PAC-Bayes-NTK uses Jacobian w.r.t. parameter. Therefore, **Construction of KL divergence** of proof of Theorem 2.2 is preserved except

$$\Sigma_{\mathbb{P}'}^{-1}\Sigma_{\mathbb{Q}'} = (\mathbf{I}_P + \frac{\alpha^2}{\sigma^2}\mathbf{J}_\theta^\top \mathbf{J}_\theta)^{-1}$$

and $\beta'_i$ are eigenvalues of $\Sigma_{\mathbb{P}'}^{-1}\Sigma_{\mathbb{Q}'}$. Note that these eigenvalues satisfy

$$\beta'_i = \frac{\sigma^2}{\sigma^2 + \alpha^2\lambda'_i}$$

where $\lambda'_i$ are eigenvalues of $\mathbf{J}_\theta^\top \mathbf{J}_\theta$. $\qquad \square$

**Remark B.2** (Function-preserving scale transformation to NTK). *In contrast to the CTK, the scale invariance property is not applicable to the NTK due to the Jacobian w.r.t. parameter:*

$$\mathbf{J}_\theta(x, \mathcal{T}(\psi)) = \frac{\partial}{\partial \mathcal{T}(\psi)} f(x, \mathcal{T}(\psi)) = \frac{\partial}{\partial \mathcal{T}(\psi)} f(x, \psi) = \mathbf{J}_\theta(x, \psi)\mathcal{T}^{-1}$$

*If we assume all parameters are scale-invariant (or equivalently masking the Jacobian for all non-scale-invariant parameters as in the proof of Proposition 3.1), the scale of NTK is proportional to the inverse scale of parameters.*

## C DETAILS OF SQUARED LOSS FOR CLASSIFICATION TASKS

For the classification tasks in Sec. 4.2, we use the squared loss instead of the cross-entropy loss since our theoretical results are built on the squared loss. Here, we describe how we use the squared loss to mimic the cross-entropy loss. There are several works (Lee et al., 2020; Hui & Belkin, 2021) that utilized the squared loss for the classification task instead of the cross-entropy loss. Specifically, Lee et al. (2020) used

$$\mathcal{L}(\mathcal{S}, \theta) = \frac{1}{2NK} \sum_{(x_i, y_i) \in \mathcal{S}} \|f(x_i, \theta) - y_i\|^2$$

where $C$ is the number of classes, and Hui & Belkin (2021) used

$$\ell((x, c), \theta) = \frac{1}{2K}\left(k(f_c(x, \theta) - M)^2 + \sum_{i=1, i \neq c}^{K} f_i(x, \theta)^2\right)$$

for single data loss, where $\ell((x, c), \theta)$ is sample loss given input $x$, target $c$ and parameter $\theta$, $f_i(x, \theta) \in \mathbb{R}$ is the $i$-th component of $f(x, \theta) \in \mathbb{R}^K$, $k$ and $M$ are dataset-specific hyper-parameters.

These works used the mean for reducing the vector-valued loss into a scalar loss. However, this can be problematic when the number of classes is large. When the number of classes increases, the denominator of the mean (the number of classes) increases while the target value remains 1 (one-hot label). As a result, the scale of a gradient for the target class becomes smaller. To avoid such an

unfavorable effect, we use the sum for reducing vector-valued loss into a scalar loss instead of taking the mean, i.e.,

$$\ell((x, c), \theta) = \frac{1}{2} \left( (f_c(x, \theta) - 1)^2 + \sum_{i=1, i \neq c}^{K} f_i(x, \theta)^2 \right)$$

This analysis is consistent with the hyper-parameter selection in Hui & Belkin (2021). They used larger $k$ and $M$ as the number of classes increases (e.g., $k = 1$, $M = 1$ for CIFAR-10 (Krizhevsky, 2009), but $k = 15$, $M = 30$ for ImageNet (Deng et al., 2009)) which results in manually compensating the scale of the gradient to the target class label.

## D  DERIVATION OF PAC-BAYES POSTERIOR

**Derivation of** $\mathbb{Q}_{\theta^*}(c)$

For Bayesian linear regression, we compute the posterior of $\beta \in \mathbb{R}^P$

$$y_i = x_i \beta + \epsilon_i, \quad \text{for } i = 1 \ldots, M$$

where $\epsilon_i \sim \mathcal{N}(0, \sigma^2)$ is i.i.d. sampled and the prior of $\beta$ is given as $\beta \sim \mathcal{N}(\mathbf{0}_P, \alpha^2 \mathbf{I}_P)$. By concatenating this, we get

$$\mathbf{y} = \mathbf{X}\beta + \varepsilon$$

where $\mathbf{y} \in \mathbb{R}^M$, $\mathbf{X} \in \mathbb{R}^{M \times p}$ are concatenation of $y_i, x_i$, respectively and $\varepsilon \sim \mathcal{N}(\mathbf{0}_M, \sigma^2 \mathbf{I}_M)$. It is well known (Bishop, 2006; Murphy, 2012) that the posterior of $\beta$ for this problem is

$$\beta \sim \mathcal{N}\left(\beta | \mu, \Sigma\right)$$

$$\mu := \frac{\Sigma \mathbf{X}^\top y}{\sigma^2}$$

$$\Sigma := \left( \frac{\mathbf{I}_P}{\alpha^2} + \frac{\mathbf{X}^\top \mathbf{X}}{\sigma^2} \right)^{-1}.$$

Similarly, we define the Bayesian linear regression problem as

$$y_i = f(x_i, \theta^*) + \mathbf{J}_\theta(x_i, \theta^*)\mathrm{diag}(\theta^*)c + \epsilon_i, \quad \text{for } i = 1 \ldots, NK$$

where $M = NK$ and the regression coefficient is $\beta = c$ in this case. Thus, we treat $y_i - f(x_i, \theta^*)$ as a target and $\mathbf{J}_\theta(x_i, \theta^*)\mathrm{diag}(\theta^*)$ as an input of linear regression problem. By concatenating this, we get

$$\mathcal{Y} = f(\mathcal{X}, \theta^*) + \mathbf{J}_c c + \varepsilon \Rightarrow (\mathcal{Y} - f(\mathcal{X}, \theta^*)) = \mathbf{J}_c c + \varepsilon.$$

By plugging this into the posterior of the Bayesian linear regression problem, we get

$$\mathbb{Q}_{\theta^*}(c) := \mathcal{N}(c | \mu_\mathbb{Q}, \Sigma_\mathbb{Q})$$

$$\mu_\mathbb{Q} := \frac{\Sigma_\mathbb{Q} \mathbf{J}_c^\top (\mathcal{Y} - f(\mathcal{X}, \theta^*))}{\sigma^2} = \frac{\Sigma_\mathbb{Q} \mathrm{diag}(\theta^*) \mathbf{J}_\theta^\top (\mathcal{Y} - f(\mathcal{X}, \theta^*))}{\sigma^2}$$

$$\Sigma_\mathbb{Q} := \left( \frac{\mathbf{I}_P}{\alpha^2} + \frac{\mathbf{J}_c^\top \mathbf{J}_c}{\sigma^2} \right)^{-1} = \left( \frac{\mathbf{I}_P}{\alpha^2} + \frac{\mathrm{diag}(\theta^*) \mathbf{J}_\theta^\top \mathbf{J}_\theta \mathrm{diag}(\theta^*)}{\sigma^2} \right)^{-1}$$

**Derivation of** $\mathbb{Q}_{\theta^*}(\psi)$ We define perturbed parameter $\psi$ as follows

$$\psi := \theta^* + \theta^* \odot c.$$

Since $\psi$ is affine to $c$, we get the distribution of $\psi$ as

$$\mathbb{Q}_{\theta^*}(\psi) := \mathcal{N}(\psi | \mu_\mathbb{Q}^\psi, \Sigma_\mathbb{Q}^\psi)$$

$$\mu_\mathbb{Q}^\psi := \theta^* + \theta^* \odot \mu_\mathbb{Q}$$

$$\Sigma_\mathbb{Q}^\psi := \mathrm{diag}(\theta^*) \Sigma_\mathbb{Q} \mathrm{diag}(\theta^*) = \left( \frac{\mathrm{diag}(\theta^*)^{-2}}{\alpha^2} + \frac{\mathbf{J}_\theta^\top \mathbf{J}_\theta}{\sigma^2} \right)^{-1}$$

# E    REPRESENTATIVE CASES OF FUNCTION-PRESERVING SCALING TRANSFORMATIONS

**Activation-wise rescaling transformation (Tsuzuku et al., 2020; Neyshabur et al., 2015a)** For NNs with positive homogeneous (e.g., ReLU) activations, following holds for $\forall x \in \mathbb{R}^d, \gamma > 0$: $f(x, \theta) = f(x, \mathcal{R}_{\gamma,l,k}(\theta))$, where rescale transformation $\mathcal{R}_{\gamma,l,k}(\cdot)^2$ is defined as

$$(\mathcal{R}_{\gamma,l,k}(\theta))_i = \begin{cases} \gamma \cdot \theta_i & \text{, if } \theta_i \in \{\text{param. subset connecting as input edges to } k\text{-th activation at } l\text{-th layer}\} \\ \theta_i/\gamma & \text{, if } \theta_i \in \{\text{param. subset connecting as output edges to } k\text{-th activation at } l\text{-th layer}\} \\ \theta_i & \text{, for } \theta_i \text{ in the other cases} \end{cases}$$

(19)

Note that $\mathcal{R}_{\gamma,l,k}(\cdot)$ is a finer-grained rescaling transformation than layer-wise rescaling transformation (i.e., common $\gamma$ for all activations in layer $l$) discussed in Dinh et al. (2017). Dinh et al. (2017) showed that even layer-wise rescaling transformations can sharpen pre-trained solutions in terms of the trace of Hessian (i.e., contradicting the FM hypothesis). This contradiction also occurs in previous PAC-Bayes bounds (Tsuzuku et al., 2020; Kwon et al., 2021) due to the scale-dependent term.

For example, let us assume a linear NN as

$$f(x, (\theta_1, \theta_2, \theta_3, \theta_4)) = (\theta_3, \theta_4)^\top \begin{pmatrix} \theta_1 \\ \theta_2 \end{pmatrix} x = \theta_1\theta_3 x + \theta_2\theta_4 x$$

Then, the Jacobian of this NN would be

$$\mathbf{J}_\theta(x, \theta) = \begin{pmatrix} \theta_3 x & \theta_4 x \\ \theta_1 x & \theta_2 x \end{pmatrix}$$

From above discussion, $(0.5, 1, 2, 1) = \mathcal{R}_{2,1,1}((1, 1, 1, 1))$ would change the Jacobian while maintaining the predictions. To show this,

$$f(x, (0.5, 1, 2, 1)) = f(x, (1, 1, 1, 1)) = x + x = 2x$$

for all $x \in \mathbb{R}$. However,

$$\mathbf{J}_\theta(x, (0.5, 1, 2, 1)) = \begin{pmatrix} 2x & x \\ 0.5x & x \end{pmatrix} \neq \begin{pmatrix} x & x \\ x & x \end{pmatrix} = \mathbf{J}_\theta(x, (1, 1, 1, 1))$$

in general. Therefore, we verified that the activation-wise rescaling transformation $\mathcal{R}_{2,1,1}$ is a valid function-preserving scale transformation.

**WD with BN layers (Ioffe & Szegedy, 2015)** For parameters $W \in \mathbb{R}^{n_l \times n_{l-1}}$ preceding BN layer,

$$\mathrm{BN}((\mathrm{diag}(\gamma)W)u) = \mathrm{BN}(Wu)$$

(20)

for an input $u \in \mathbb{R}^{n_{l-1}}$ and a positive vector $\gamma \in \mathbb{R}_+^{n_l}$. This implies that scaling transformations on these parameters preserve function represented by NNs for $\forall x \in \mathbb{R}^d, \gamma \in \mathbb{R}_+^{n_l}$: $f(x, \theta) = f(x, \mathcal{S}_{\gamma,l,k}(\theta))$, where scaling transformation $\mathcal{S}_{\gamma,l,k}(\cdot)$ is defined for $i = 1, \ldots, P$

$$(\mathcal{S}_{\gamma,l,k}(\theta))_i = \begin{cases} \gamma_k \cdot \theta_i & \text{, if } \theta_i \in \{\text{param. subset connecting as input edges to } k\text{-th activation at } l\text{-th layer}\} \\ \theta_i & \text{, for } \theta_i \text{ in the other cases} \end{cases}$$

(21)

Note that the WD (Loshchilov & Hutter, 2019; Zhang et al., 2019) can be implemented as a realization of $\mathcal{S}_{\gamma,l,k}(\cdot)$ (e.g., $\gamma = 0.9995$ for all activations preceding BN layers). Therefore, thanks to Proposition 2.1 and Theorem 2.4, our CTK-based bound is invariant to WD applied to parameters before BN layers. We also refer to (Van Laarhoven, 2017; Lobacheva et al., 2021) for the optimization perspective of WD with BN.

For example, let us assume a BN layer with single activation as $\mathrm{BN}(\theta x)$ where $x, \theta \in \mathbb{R}$ and batch of input is given as $0, 0, 2, 2$. Then, the Jacobian of NN would be

$$\mathbf{J}_\theta(x, \theta) = \frac{x}{|\theta|}$$

---

[2] For a simple two layer linear NN $f(x) := W_2\sigma(W_1 x)$ with weight matrix $W_1, W_2$, the first case of equation 19 corresponds to $k$-th row of $W_1$ and the second case of equation 19 corresponds to $k$-th column of $W_2$.

as $f(x, \theta) = \frac{\theta x}{|\theta|}$ and the denominator is detached from backpropagation computation for auto-differentiation packages (e.g., TensorFlow, Pytorch, and JAX). From the above discussion, $0.9995 = \mathcal{S}_{0.9995,1,1}(1)$ would change the Jacobian while maintaining the predictions. To show this,

$$f(x, 1) = f(x, 0.9995) = x$$

for all inputs. On the other hand,

$$\mathbf{J}_\theta(x, 0.9995) = \frac{x}{0.9995} \neq x = \mathbf{J}_\theta(x, 1).$$

Therefore, we showed that the WD with BN layer $\mathcal{S}_{0.9995,1,1}$ is a valid function-preserving scale transformation.

# F    IMPLEMENTATIONS

## F.1    RTO IMPLEMENTATION OF CONNECTIVITY LAPLACE

---
**Algorithm 1** Connectivity Laplace RTO

---
**Require:** training data $\mathcal{S}$, pre-trained parameter $\theta^*$, number of samples $M$, prior scale $\alpha$, observational noise scale $\sigma$
    **for** $m = 1, \ldots, M$ **do**
        Sample parameter noise $c_0^m \sim \mathcal{N}(c_0 | \mathbf{0}_P, \alpha^2 \mathbf{I}_P)$ and label noise $\varepsilon^m \sim \mathcal{N}(\varepsilon | \mathbf{0}_{NK}, \sigma^2 \mathbf{I}_{NK})$
        **for** $t = 1, \ldots, T$ **do**
            Randomly draw a mini-batch $\mathcal{B}$ from $\mathcal{S}$.
            Define $g_{\theta^*}^{\text{lin}}(\mathcal{X}_\mathcal{B}, c^m) = f(\mathcal{X}_\mathcal{B}, \theta^*) + \mathbf{J}_\theta(\mathcal{X}_\mathcal{B}, \theta^*)\text{diag}(\theta^*)c^m$ for mini-batch input $\mathcal{X}_\mathcal{B}$.
            Define $L(c^m) = \frac{1}{2|\mathcal{B}|\sigma^2}\|\mathcal{Y}_\mathcal{B} + \varepsilon_\mathcal{B}^m - g_{\theta^*}^{\text{lin}}(\mathcal{X}_\mathcal{B}, c^m)\|_2^2 + \frac{1}{2|\mathcal{B}|\alpha^2}\|c^m - c_0^m\|_2^2$.
            Compute backpropagation of $L(c^m)$ w.r.t. connectivity $c^m$.
            Update $c^m$ with SGD optimizer.
        **end for**
    **end for**
    **return** Samples from Connectivity Laplace $\{c^m\}_{m=1}^M$.

---

To estimate the empirical/generalization bound in Sec. 2.4 and calibrate uncertainty in Sec. 4.2, we need to sample $c$ from the posterior $\mathbb{Q}_{\theta^*}(c)$. For this, we sample perturbations $\delta$ in connectivity space

$$\delta \sim \mathcal{N}\left(\delta | \mathbf{0}_P, \left(\frac{\mathbf{I}_P}{\alpha^2} + \frac{\mathbf{J}_c^\top \mathbf{J}_c}{\sigma^2}\right)^{-1}\right)$$

so that $c = \mu_\mathbb{Q} + \delta$ for equation 6. To sample this, we provide a novel approach to sample from LA/CL without curvature approximation. To this end, we consider the following optimization problem

$$\arg\min_c L(c) := \arg\min_c \frac{1}{2N\sigma^2}\|\mathcal{Y} - f(\mathcal{X}, \theta^*) - \mathbf{J}_c c + \varepsilon\|^2 + \frac{1}{2N\alpha^2}\|c - c_0\|_2^2$$

where $\varepsilon \sim \mathcal{N}(\varepsilon | \mathbf{0}_{NK}, \sigma^2 \mathbf{I}_{NK})$ and $c_0 \sim \mathcal{N}(c_0 | \mathbf{0}_P, \alpha^2 \mathbf{I}_P)$. By first-order optimality condition, we have

$$N\nabla_c L(c) = -\frac{\mathbf{J}_c^\top(\mathcal{Y} - f(\mathcal{X}, \theta^*) - \mathbf{J}_c c^* + \varepsilon)}{\sigma^2} + \frac{c^* - c_0}{\alpha^2} = \mathbf{0}_P.$$

By arranging this w.r.t. optimizer $c^*$, we get

$$c^* = \left(\mathbf{J}_c^\top \mathbf{J}_c + \frac{\sigma^2}{\alpha^2}\mathbf{I}_P\right)^{-1}\left(\mathbf{J}_c^\top(\mathcal{Y} - f(\mathcal{X}, \theta^*)) + \frac{\sigma^2}{\alpha^2}c_0 + \mathbf{J}_c \varepsilon\right)$$

$$= \left(\mathbf{J}_c^\top \mathbf{J}_c + \frac{\sigma^2}{\alpha^2}\mathbf{I}_P\right)^{-1}\mathbf{J}_c^\top(\mathcal{Y} - f(\mathcal{X}, \theta^*)) + \left(\mathbf{J}_c^\top \mathbf{J}_c + \frac{\sigma^2}{\alpha^2}\mathbf{I}_P\right)^{-1}\left(\frac{\sigma^2}{\alpha^2}c_0 + \mathbf{J}_c \varepsilon\right)$$

$$= \underbrace{\left(\frac{\mathbf{I}_P}{\alpha^2} + \frac{\mathbf{J}_c^\top \mathbf{J}_c}{\sigma^2}\right)^{-1}\frac{\mathbf{J}_c^\top(\mathcal{Y} - f(\mathcal{X}, \theta^*))}{\sigma^2}}_{\text{Deterministic}} + \underbrace{\left(\frac{\mathbf{I}_P}{\alpha^2} + \frac{\mathbf{J}_c^\top \mathbf{J}_c}{\sigma^2}\right)^{-1}\left(\frac{c_0}{\alpha^2} + \frac{\mathbf{J}_c^\top \varepsilon}{\sigma^2}\right)}_{\text{Stochastic}}$$

Since both $\varepsilon$ and $c_0$ are sampled from independent Gaussian distributions, we have

$$z := \left( \frac{c_0}{\alpha^2} + \frac{\mathbf{J}_c^\top \varepsilon}{\sigma^2} \right) \sim \mathcal{N}\left( z | \mathbf{0}_P, \frac{\mathbf{I}_P}{\alpha^2} + \frac{\mathbf{J}_c^\top \mathbf{J}_c}{\sigma^2} \right)$$

Therefore, optimal solution of randomized optimization problem $\arg\min_c L(c)$ is

$$c \sim \mathcal{N}\left( c \,\middle|\, \left( \frac{\mathbf{I}_P}{\alpha^2} + \frac{\mathbf{J}_c^\top \mathbf{J}_c}{\sigma^2} \right)^{-1} \frac{\mathbf{J}_c^\top (\mathcal{Y} - f(\mathcal{X}, \theta^*))}{\sigma^2}, \left( \frac{\mathbf{I}_P}{\alpha^2} + \frac{\mathbf{J}_c^\top \mathbf{J}_c}{\sigma^2} \right)^{-1} \right) = \mathcal{N}(c | \mu_\mathbb{Q}, \Sigma_\mathbb{Q})$$

Similarly, sampling from CL can be implemented as the following optimization problem.

$$\arg\min_c L(c) := \arg\min_c \frac{1}{2N\sigma^2} \|\mathbf{J}_c c - \varepsilon\|^2 + \frac{1}{2N\alpha^2} \|c - c_0\|_2^2$$

where $\varepsilon \sim \mathcal{N}(\varepsilon | \mathbf{0}_{NK}, \sigma^2 \mathbf{I}_{NK})$ and $c_0 \sim \mathcal{N}(c_0 | \mathbf{0}_P, \alpha^2 \mathbf{I}_P)$. Since we sample the noise of data/perturbation and optimize the perturbation, this can be interpreted as a Randomize-Then-Optimize implementation of Laplace approximation and Connectivity Laplace (Bardsley et al., 2014; Matthews et al., 2017). In Algorithm 1, we provide a pseudo-code for the RTO implementation of CL. Note that both time and memory complexity of computing linearized NN for mini-batch $\mathcal{B}$ is comparable to a forward propagation as shown in Novak et al. (2022) with `jax.jvp` function in JAX (Bradbury et al., 2018). Therefore, the time/memory complexity of mini-batch JVP would be $\mathcal{O}(|\mathcal{B}|LW^2)/\mathcal{O}(|\mathcal{B}|W + LW^2 + NK)$ for MLPs with width $W$ and depth $L$.

### F.2 Computing Connectivity Sharpness

---
**Algorithm 2** Hutchison's method for computing Connectivity sharpness

---
**Require:** training data $\mathcal{S}$, pre-trained parameter $\theta^*$, number of samples $M$
    $x_\mathcal{B} = 0$
    **for** $m = 1, \dots, M$ **do**
        Sample $z^m \sim \mathcal{N}(\varepsilon | \mathbf{0}_{NK}, \sigma^2 \mathbf{I}_{NK})$
        **for** $t = 1, \dots, T$ **do**
            Sequentially draw a mini-batch $\mathcal{B}$ from $\mathcal{S}$.
            Compute vector-Jacobian product $v_\mathcal{B}^m = z_\mathcal{B}^m \mathbf{J}_\theta(\mathcal{X}, \theta^*)\mathrm{diag}(\theta^*)$.
            Compute $x_\mathcal{B} = x_\mathcal{B} + \|v_\mathcal{B}^m\|_2^2/T$
        **end for**
    **end for**
    **return** Estimated Connectivity Sharpness $x_\mathcal{B}$

---

It is well known that empirical NTK or Jacobian is intractable in the modern architecture of NNs (e.g., ResNet (He et al., 2016) or BERT (Devlin et al., 2018)). Therefore, one might wonder how Connectivity Sharpness (CS) can be computed for these architectures. One can compute CS with Hutchison's method (Hutchinson, 1989; Meyer et al., 2021) as it is defined as a trace of empirical CTK. According to Hutchison's method, trace of a matrix $A \in \mathbb{R}^{m \times m}$ is

$$\mathrm{tr}(A) = \mathrm{tr}(A\mathbf{I}_p) = \mathrm{tr}(A\mathbb{E}_z[zz^\top]) = \mathbb{E}_z[\mathrm{tr}(Azz^\top)] = \mathbb{E}_z[\mathrm{tr}(z^\top Az)] = \mathbb{E}_z[z^\top Az]$$

where $z \in \mathbb{R}^m$ is a random variable with $\mathrm{cov}(z) = \mathbf{I}_m$ (e.g., standard normal distribution or Rademacher distribution). Since $A = \mathbf{C}_\mathcal{X}^{\theta^*} = \mathbf{J}_c \mathbf{J}_c^\top \in \mathbb{R}^{NK}$ in our case, we further use mini-batch approximation to compute $z^\top Az$: (i) Sample $z_\mathcal{B}^m \in \mathbb{R}^{MK}$ from Rademacher distribution for mini-batch $\mathcal{M}$ with size $M$ and (ii) compute $v_\mathcal{B}^m := \mathbf{J}_c(\mathcal{X}_\mathcal{B}, \mathbf{0}_p)^\top z_\mathcal{B}^m \in \mathbb{R}^P$ with vector-Jacobian product of JAX (Bradbury et al., 2018) (or it can simply computed using standard backprop) and (iii) compute $x_\mathcal{B}^m = \|v_\mathcal{B}^m\|_2^2$. Then, the sum of $x_\mathcal{B}^m$ for all mini-batch in the training dataset is a Monte-Carlo approximation of CS with sample size 1. Empirically, we found that this approximation is sufficiently stable to capture the correlation between sharpness and generalization as shown in Sec. 4.1. In Algorithm 2, we provide a pseudo-code for the implementation. Note that both time and memory complexity of computing $v_\mathcal{B}^m$ is comparable to a backward propagation as shown in Novak et al. (2022) with `jax.vjp` function in JAX (Bradbury et al., 2018). Therefore, the time/memory complexity of mini-batch VJP would be $\mathcal{O}(|\mathcal{B}|LW^2)/\mathcal{O}(|\mathcal{B}|LW + LW^2 + NKW)$ for MLPs with width $W$ and depth $L$.

## G   PREDICTIVE UNCERTAINTY OF CONNECTIVITY/LINEARIZED LAPLACE

In this section, we derive predictive uncertainty of Linearized Laplace (LL) and Connectivity Laplace (CL). By matrix inversion lemma (Woodbury, 1950), the weight covariance of LL is

$$(\mathbf{I}_p/\alpha^2 + \mathbf{J}_\theta(\mathcal{X}, \theta^*)^\top \mathbf{J}_\theta(\mathcal{X}, \theta^*)/\sigma^2)^{-1} = \alpha^2 \mathbf{I}_p - \alpha^2 \mathbf{J}_\theta(\mathcal{X}, \theta^*)^\top (\frac{\sigma^2}{\alpha^2}\mathbf{I}_{Nk} + \Theta_{\mathcal{X}\mathcal{X}}^{\theta^*})^{-1}\mathbf{J}_\theta(\mathcal{X}, \theta^*).$$

Therefore, if $\sigma^2/\alpha^2 \to 0$, then the weight covariance of LL converges to

$$\alpha^2 \mathbf{I}_p - \alpha^2 \mathbf{J}_\theta(\mathcal{X}, \theta^*)^\top \Theta_{\mathcal{X}\mathcal{X}}^{\theta^* -1}\mathbf{J}_\theta(\mathcal{X}, \theta^*).$$

With this weight covariance and linearized NN, the predictive uncertainty of LL is

$$f_{\theta^*}^{\text{lin}}(x, \theta)|p_{\text{LA}}(\theta|\mathcal{S}) \sim \mathcal{N}(f(x, \theta^*), \alpha^2 \Theta_{xx}^{\theta^*} - \alpha^2 \Theta_{x\mathcal{X}}^{\theta^*}\Theta_{\mathcal{X}\mathcal{X}}^{\theta^* -1}\Theta_{\mathcal{X}x}^{\theta^*}).$$

Similarly, the predictive uncertainty of CL is

$$f_{\theta^*}^{\text{lin}}(x, \theta)|\mathbb{Q}_{\theta^*}(\theta) \sim \mathcal{N}(f(x, \theta^*), \alpha^2 \mathbf{C}_{xx}^{\theta^*} - \alpha^2 \mathbf{C}_{x\mathcal{X}}^{\theta^*}\mathbf{C}_{\mathcal{X}\mathcal{X}}^{\theta^* -1}\mathbf{C}_{\mathcal{X}x}^{\theta^*}).$$

## H   DETAILED EXPERIMENTAL SETTINGS

### H.1   BOUND ESTIMATION

We pre-train ResNet-18 (He et al., 2016) with a mini-batch size of 1K on $\mathcal{S}_\mathbb{P}$ with SGD of initial learning rate 0.4 and momentum 0.9. We use cosine annealing for learning rate scheduling (Loshchilov & Hutter, 2016) with a warmup for the initial 10% training step. We fix $\delta = 0.1$, $\alpha = 0.1$, and $\sigma = 1.0$ to compute equation 8. These values are chosen so that the PAC-Bayes-CTK and NTK bounds fall within the non-vacuous range. We use 3 random seeds to compute the standard errors. Additional results on few pre-training data with $N_\mathbb{P} = 5,000$ and $N_\mathbb{Q} = 45,000$ are presented in Appendix I.

### H.2   CONNECTIVITY SHARPNESS

Table 6: Configuration of hyper-parameter

| | |
|---|---|
| network depth | 1, 2, 3 |
| network width | 32, 64, 128 |
| learning rate | 0.1, 0.032, 0.001 |
| WD | 0.0, 1e-4, 5e-4 |
| mini-batch size | 256, 1024, 4096 |

To verify that the CS has a better correlation with generalization performance compared to existing sharpness measures, we evaluate the three metrics: (a) Kendall's rank-correlation coefficient $\tau$ (Kendall, 1938) which considers the consistency of a sharpness measure with generalization gap (i.e., if one has higher sharpness, then so has higher generalization gap) (b) granulated Kendall's coefficient (Jiang et al., 2020) which examines Kendall's rank-correlation coefficient w.r.t. individual hyper-parameters to separately evaluate the effect of each hyper-parameter to generalization gap (c) conditional independence test (Jiang et al., 2020) which captures the causal relationship between measure and generalization.

Three metrics are compared with the following baselines: trace of Hessian ($\text{tr}(\mathbf{H})$; (Keskar et al., 2017)), trace of Fisher information matrix ($\text{tr}(\mathbf{F})$; (Jastrzebski et al., 2021)), trace of empirical NTK at $\theta^*$ ($\text{tr}(\Theta^{\theta^*})$), and four PAC-Bayes bound based measures, sharpness-orig (SO), PAC-Bayes-orig (PO), $1/\alpha'$ sharpness mag (SM), and $1/\sigma'$ PAC-Bayes mag (PM), which are eq. (52), (49), (62), (61) in Jiang et al. (2020).

For the granulated Kendall's coefficient, we use 5 hyper-parameters: network depth, network width, learning rate, WD and mini-batch size, along with 3 options for each hyper-parameters as in Table 6. We use the VGG-13 as a base model and we adjust the depth and width of each conv block. We add

Table 7: Example of network configuration with respect to the depth 1, width 128 in (Simonyan & Zisserman, 2015)-style.

| ConvNet Configuration |
|:---:|
| input (224 × 224 RGB image) |
| Conv3-128 
 BN 
 ReLU |
| MaxPool |
| Conv3-256 
 BN 
 ReLU |
| MaxPool |
| Conv3-512 
 BN 
 ReLU |
| MaxPool |
| Conv3-1024 
 BN 
 ReLU |
| MaxPool |
| Conv3-1024 
 BN 
 ReLU |
| MaxPool |
| FC-4096 
 ReLU |
| FC-4096 
 ReLU |
| FC-1000 |

BN layers after the convolution layer for each block. Specifically, the number of convolution layers of each conv block is the depth and the number of channels of convolution layers of the first conv block is the width. For the subsequent conv blocks, we follow the original VGG width multipliers ($\times 2$, $\times 4$, $\times 8$). An example with depth 1 and width 128 is depicted in Table 7.

We use an SGD optimizer with momentum of 0.9. We train each model for 200 epochs and use cosine learning rate scheduler (Loshchilov & Hutter, 2016) with 30% of initial epochs as warm-up epochs. The standard data augmentations (padding, random crop, random horizontal flip, and normalization) for CIFAR-10 are used for training data. For the analysis, we only use models with above 99% training accuracy following Jiang et al. (2020). As a result, we use 200 out of 243 trained models for our correlation analysis. For every experiment, we use 8 NVIDIA RTX 3090 GPUs.

### H.3    BNN EXPERIMENTS

**UCI regression** We train MLP with a single hidden layer. We fix $\sigma = 1$ and choose $\alpha$ from {0.01, 0.1, 1, 10, 100} using log-likelihood of validation dataset since the optimal $\alpha$ varies for each regression dataset. We use 8 random seeds to compute the average and standard error of the test negative log-likelihoods.

**Image classification task** We pre-train models for 200 epochs CIFAR-10/100 dataset (Krizhevsky, 2009) with ResNet-18(He et al., 2016) as mentioned in Section 2.4. We choose ensemble size $M$ as 8

except Deep Ensemble (Lakshminarayanan et al., 2017) and Batch Ensemble (Wen et al., 2020). We use 4 ensemble members for Deep Ensemble and Batch Ensemble due to computational cost. We used 4 random seeds to compute the standard errors except for Deep Ensemble, which ensembles the NNs from 4 different random seeds.

For evaluation, we define a prediction of single-member as the one-hot representation of network output with label smoothing. We select the label smoothing coefficient as 0.01 for CIFAR-10 and 0.1 for CIFAR-100. We define ensemble prediction as an averaged prediction of single-member predictions. For OOD detection, we use the variance of prediction in output space, which is competitive to recent OOD detection methods (Ren et al., 2019; Van Amersfoort et al., 2020). We use 0.01 for $\sigma$ and select the best $\alpha$ with cross-validation. For every experiment, we used 8 NVIDIA RTX 3090 GPUs.

# I    ADDITIONAL RESULTS ON BOUND ESTIMATION

Table 8: Results for experiments on PAC-Bayes-CTK/NTK estimation with $N_{\mathbb{P}} = 5,000$ and $N_{\mathbb{Q}} = 45,000$. We use 4 random seeds to compute error bars.

| **CIFAR-10** | | PAC-Bayes-CTK | | | | PAC-Bayes-NTK | | |
|---|---|---|---|---|---|---|---|---|
| Parameter scale | 0.5 | 1.0 | 2.0 | 4.0 | 0.5 | 1.0 | 2.0 | 4.0 |
| Trace ($\times 10^{-4}$) | 1.30 ± 0.13 | 1.31 ± 0.13 | 1.31 ± 0.13 | 1.31 ± 0.13 | 541.36 ± 32.58 | 139.50 ± 9.06 | 54.50 ± 0.63 | 48.86 ± 0.90 |
| Perturbation | 335.02 ± 9.57 | 338.85 ± 9.39 | 337.21 ± 9.47 | 335.87 ± 8.39 | 202.53 ± 3.14 | 315.99 ± 2.75 | 447.97 ± 3.75 | 549.35 ± 7.64 |
| Sharpness | 13.48 ± 0.63 | 13.49 ± 0.63 | 13.49 ± 0.63 | 13.49 ± 0.63 | 64.70 ± 0.55 | 52.55 ± 0.58 | 44.25 ± 0.11 | 42.30 ± 0.20 |
| KL | 174.25 ± 5.03 | 176.17 ± 4.94 | 175.35 ± 4.99 | 174.68 ± 4.46 | 133.61 ± 1.48 | 184.27 ± 1.24 | 246.11 ± 1.86 | 295.83 ± 3.82 |
| Test err. ($\times 10^2$) | 23.75 ± 0.17 | 23.87 ± 0.19 | 23.94 ± 0.22 | 23.94 ± 0.22 | 31.33 ± 0.30 | 27.99 ± 0.20 | 26.15 ± 0.22 | 25.64 ± 0.20 |
| Bound ($\times 10^2$) | 27.71 ± 0.15 | 27.82 ± 0.17 | 27.84 ± 0.17 | 27.85 ± 0.17 | 33.59 ± 0.25 | 30.59 ± 0.10 | 29.66 ± 0.14 | 29.98 ± 0.18 |
| **CIFAR-100** | | PAC-Bayes-CTK | | | | PAC-Bayes-NTK | | |
| Parameter scale | 0.5 | 1.0 | 2.0 | 4.0 | 0.5 | 1.0 | 2.0 | 4.0 |
| Trace ($\times 10^{-4}$) | 2.22 ± 0.14 | 2.22 ± 0.14 | 2.22 ± 0.14 | 2.22 ± 0.14 | 655.76 ± 23.48 | 167.02 ± 5.78 | 57.01 ± 3.92 | 50.57 ± 4.71 |
| Perturbation | 447.86 ± 12.50 | 447.86 ± 8.47 | 445.76 ± 7.75 | 446.95 ± 8.08 | 399.66 ± 10.44 | 622.17 ± 27.92 | 790.63 ± 46.70 | 823.06 ± 55.58 |
| Sharpness | 17.11 ± 0.40 | 17.10 ± 0.40 | 17.11 ± 0.40 | 17.11 ± 0.40 | 66.36 ± 0.27 | 54.13 ± 0.25 | 44.47 ± 0.59 | 42.64 ± 0.79 |
| KL | 232.49 ± 6.18 | 232.48 ± 4.16 | 231.43 ± 3.79 | 232.03 ± 3.94 | 233.01 ± 5.26 | 338.15 ± 13.96 | 417.55 ± 23.06 | 432.85 ± 27.41 |
| Test err. ($\times 10^2$) | 65.07 ± 0.33 | 65.15 ± 0.38 | 65.21 ± 0.39 | 65.22 ± 0.39 | 69.92 ± 0.50 | 69.03 ± 0.38 | 68.37 ± 0.49 | 68.63 ± 0.50 |
| Bound ($\times 10^2$) | 70.98 ± 0.35 | 71.04 ± 0.38 | 71.07 ± 0.38 | 71.10 ± 0.39 | 71.10 ± 0.27 | 69.71 ± 0.59 | 71.44 ± 0.62 | 73.61 ± 0.49 |

# J    ADDITIONAL RESULTS ON IMAGE CLASSIFICATION

Table 9: Uncertainty calibration results on CIFAR-10 (Krizhevsky, 2009) for ResNet-18 (He et al., 2016).

| | CIFAR-10 | | | |
|---|---|---|---|---|
| | NLL (↓) | ECE (↓) | Brier. (↓) | AUC (↑) |
| Deterministic | 0.3135 ± 0.0088 | 0.0350 ± 0.0014 | 0.0875 ± 0.0026 | - |
| MCDO | 0.2845 ± 0.0148 | 0.0157 ± 0.0012 | 0.1056 ± 0.0062 | 0.9172 ± 0.0074 |
| MCBN | 0.2922 ± 0.0074 | 0.0325 ± 0.0010 | 0.0838 ± 0.0022 | 0.9431 ± 0.0033 |
| Batch Ensemble | 0.2740 ± 0.0030 | 0.0286 ± 0.0005 | 0.0814 ± 0.0009 | 0.9376 ± 0.0036 |
| Deep Ensemble | 0.1753 | 0.0067 | 0.0594 | 0.8527 |
| Linearized Laplace | 0.2077 ± 0.0032 | 0.0180 ± 0.0009 | 0.0816 ± 0.0010 | 0.8991 ± 0.0198 |
| Connectivity Laplace (Ours) | 0.2089 ± 0.0023 | 0.0120 ± 0.0019 | 0.0720 ± 0.0011 | 0.9705 ± 0.0098 |

Table 10: Uncertainty calibration results on CIFAR-10 (Krizhevsky, 2009) for VGG-13 (Simonyan & Zisserman, 2015).

| | CIFAR-10 | | | |
|---|---|---|---|---|
| | NLL (↓) | ECE (↓) | Brier. (↓) | AUC (↑) |
| Deterministic | 0.4086 ± 0.0018 | 0.0490 ± 0.0003 | 0.1147 ± 0.0005 | - |
| MCDO | 0.3889 ± 0.0049 | 0.0465 ± 0.0009 | 0.1106 ± 0.0015 | 0.7765 ± 0.0221 |
| MCBN | 0.3852 ± 0.0012 | 0.0462 ± 0.0002 | 0.1108 ± 0.0003 | 0.9051 ± 0.0065 |
| Batch Ensemble | 0.3544 ± 0.0036 | 0.0399 ± 0.0009 | 0.1064 ± 0.0012 | 0.9067 ± 0.0030 |
| Deep Ensemble | 0.2243 | 0.0121 | 0.0776 | 0.7706 |
| Linearized Laplace | 0.3366 ± 0.0013 | 0.0398 ± 0.0004 | 0.1035 ± 0.0003 | 0.8883 ± 0.0017 |
| Connectivity Laplace (Ours) | 0.2674 ± 0.0028 | 0.0234 ± 0.0011 | 0.0946 ± 0.0010 | 0.9002 ± 0.0033 |

Table 11: Uncertainty calibration results on CIFAR-100 (Krizhevsky, 2009) for VGG-13 (Simonyan & Zisserman, 2015).

| | CIFAR-100 | | | |
|---|---|---|---|---|
| | NLL ($\downarrow$) | ECE ($\downarrow$) | Brier. ($\downarrow$) | AUC ($\uparrow$) |
| Deterministic | 1.8286 ± 0.0066 | 0.1544 ± 0.0010 | 0.4661 ± 0.0018 | - |
| MCDO | 1.7439 ± 0.0089 | 0.1363 ± 0.0008 | 0.4456 ± 0.0017 | 0.6424 ± 0.0099 |
| MCBN | 1.7491 ± 0.0075 | 0.1399 ± 0.0010 | 0.4488 ± 0.0015 | 0.7039 ± 0.0197 |
| Batch Ensemble | 1.6142 ± 0.0101 | 0.1077 ± 0.0020 | 0.4143 ± 0.0027 | 0.7232 ± 0.0021 |
| Deep Ensemble | 1.2006 | 0.0456 | 0.3228 | 0.6929 |
| Linearized Laplace | 1.5806 ± 0.0054 | 0.1036 ± 0.0004 | 0.4127 ± 0.0010 | 0.6893 ± 0.0221 |
| Connectivity Laplace (Ours) | 1.4073 ± 0.0039 | 0.0703 ± 0.0028 | 0.3827 ± 0.0012 | 0.7254 ± 0.0136 |

## K  TRANSPOSED TABLE 3

Table 12: Transposed Table

| | | boston_housing | concrete_strength | energy_efficiency | kin8nm | naval_propulsion | power_plant | protein_structure | wine | yacht_hydrodynamics |
|---|---|---|---|---|---|---|---|---|---|---|
| Original | Deep Ensemble | 2.90 ± 0.03 | 3.06 ± 0.01 | 0.74 ± 0.01 | -1.07 ± 0.00 | -4.83 ± 0.00 | 2.81 ± 0.00 | 2.89 ± 0.00 | 1.21 ± 0.00 | 1.26 ± 0.04 |
| | MCDO | 2.63 ± 0.01 | 3.20 ± 0.00 | 1.92 ± 0.01 | -0.80 ± 0.01 | -3.85 ± 0.00 | 2.91 ± 0.00 | 2.96 ± 0.00 | 0.96 ± 0.01 | 2.17 ± 0.06 |
| | LL | **2.85 ± 0.01** | 3.22 ± 0.01 | 2.12 ± 0.01 | -0.90 ± 0.00 | -4.57 ± 0.00 | 2.91 ± 0.00 | 2.91 ± 0.00 | **1.24 ± 0.01** | **1.20 ± 0.04** |
| | CL | 2.88 ± 0.02 | **3.11 ± 0.02** | **0.83 ± 0.01** | **-1.07 ± 0.00** | **-4.76 ± 0.00** | **2.81 ± 0.00** | **2.89 ± 0.00** | 1.27 ± 0.01 | 1.25 ± 0.04 |
| GAP variants | Deep Ensemble | 2.71 ± 0.01 | 4.03 ± 0.07 | 0.77 ± 0.01 | -0.94 ± 0.00 | -2.22 ± 0.33 | 2.91 ± 0.00 | 3.11 ± 0.00 | 1.48 ± 0.01 | 1.71 ± 0.03 |
| | MCDO | 2.68 ± 0.01 | 3.42 ± 0.00 | 1.78 ± 0.01 | -0.71 ± 0.00 | -3.36 ± 0.01 | 2.97 ± 0.00 | 3.07 ± 0.00 | 1.03 ± 0.00 | 3.06 ± 0.02 |
| | LL | **2.74 ± 0.01** | **3.47 ± 0.01** | 2.02 ± 0.01 | -0.87 ± 0.00 | -3.66 ± 0.11 | 2.98 ± 0.00 | **3.07 ± 0.00** | 1.45 ± 0.01 | 1.78 ± 0.02 |
| | CL | 2.75 ± 0.01 | 4.03 ± 0.02 | 0.90 ± 0.02 | **-0.93 ± 0.00** | **-3.80 ± 0.07** | **2.91 ± 0.00** | 3.13 ± 0.00 | **1.43 ± 0.00** | **1.74 ± 0.01** |

Table 13: First 5 columns of transposed table

| | | boston_housing | concrete_strength | energy_efficiency | kin8nm | naval_propulsion |
|---|---|---|---|---|---|---|
| Original | Deep Ensemble | 2.90 ± 0.03 | 3.06 ± 0.01 | 0.74 ± 0.01 | -1.07 ± 0.00 | -4.83 ± 0.00 |
| | MCDO | 2.63 ± 0.01 | 3.20 ± 0.00 | 1.92 ± 0.01 | -0.80 ± 0.01 | -3.85 ± 0.00 |
| | LL | **2.85 ± 0.01** | 3.22 ± 0.01 | 2.12 ± 0.01 | -0.90 ± 0.00 | -4.57 ± 0.00 |
| | CL | 2.88 ± 0.02 | **3.11 ± 0.02** | **0.83 ± 0.01** | **-1.07 ± 0.00** | **-4.76 ± 0.00** |
| GAP variants | Deep Ensemble | 2.71 ± 0.01 | 4.03 ± 0.07 | 0.77 ± 0.01 | -0.94 ± 0.00 | -2.22 ± 0.33 |
| | MCDO | 2.68 ± 0.01 | 3.42 ± 0.00 | 1.78 ± 0.01 | -0.71 ± 0.00 | -3.36 ± 0.01 |
| | LL | **2.74 ± 0.01** | **3.47 ± 0.01** | 2.02 ± 0.01 | -0.87 ± 0.00 | -3.66 ± 0.11 |
| | CL | 2.75 ± 0.01 | 4.03 ± 0.02 | 0.90 ± 0.02 | **-0.93 ± 0.00** | **-3.80 ± 0.07** |

Table 14: Last 4 columns of transposed table

| | | power_plant | protein_structure | wine | yacht_hydrodynamics |
|---|---|---|---|---|---|
| Original | Deep Ensemble | 2.81 ± 0.00 | 2.89 ± 0.00 | 1.21 ± 0.00 | 1.26 ± 0.04 |
| | MCDO | 2.91 ± 0.00 | 2.96 ± 0.00 | 0.96 ± 0.01 | 2.17 ± 0.06 |
| | LL | 2.91 ± 0.00 | 2.91 ± 0.00 | **1.24 ± 0.01** | **1.20 ± 0.04** |
| | CL | **2.81 ± 0.00** | **2.89 ± 0.00** | 1.27 ± 0.01 | 1.25 ± 0.04 |
| GAP variants | Deep Ensemble | 2.91 ± 0.00 | 3.11 ± 0.00 | 1.48 ± 0.01 | 1.71 ± 0.03 |
| | MCDO | 2.97 ± 0.00 | 3.07 ± 0.00 | 1.03 ± 0.00 | 3.06 ± 0.02 |
| | LL | 2.98 ± 0.00 | **3.07 ± 0.00** | 1.45 ± 0.01 | 1.78 ± 0.02 |
| | CL | **2.91 ± 0.00** | 3.13 ± 0.00 | **1.43 ± 0.00** | **1.74 ± 0.01** |

## L  EIGENSPECTRUMS OF EMPIRICAL CTK AND NTK

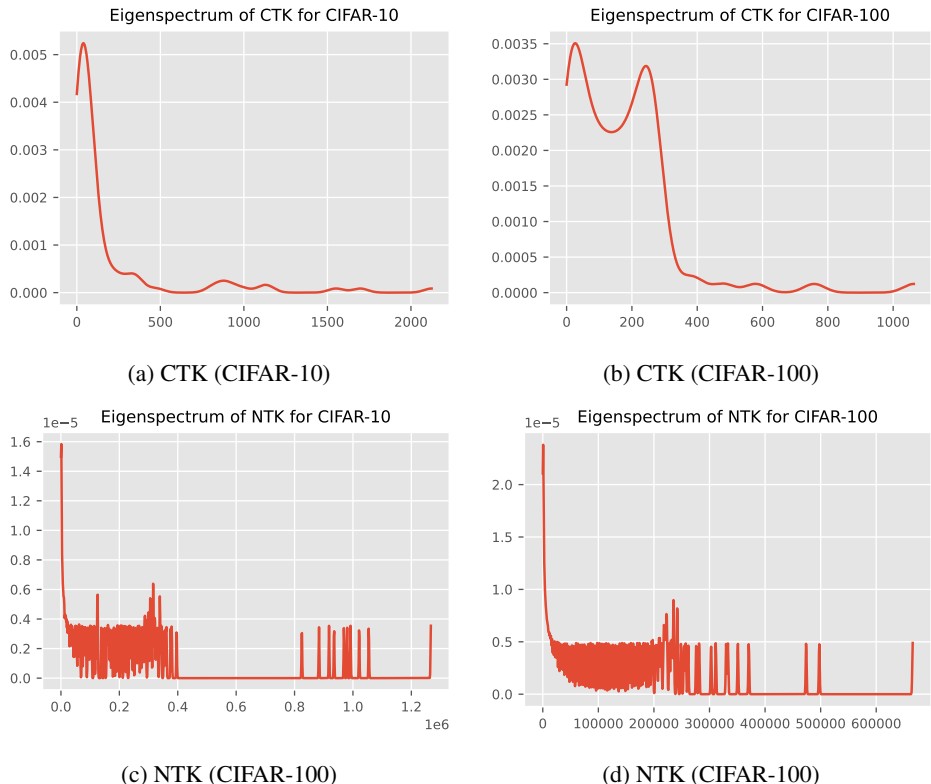

(a) CTK (CIFAR-10)        (b) CTK (CIFAR-100)

(c) NTK (CIFAR-100)        (d) NTK (CIFAR-100)

Figure 3: Eigenspectrums of CTK and NTK for CIFAR datasets

In this section, we follow Algorithm 1 of Ghorbani et al. (2019) to visualize the eigenvalue densities of empirical CTK and NTK. We use 100 Lanczos iterations (Appendix F) with 4 realizations and fix the bandwidth of the RBF kernel as the difference between the maximum and minimum eigenvalue following the implementation of Ghorbani et al. (2019). We present the results in Figure 3.

Figure 3 shows that empirical CTKs have positively skewed (i.e., right-tailed) eigenspectrums with modes close to 0. In other words, many of the non-zero eigenvalues of CTK are close to zero. Therefore, the corresponding $h(\beta_i)$ for these eigenvalues are also close to zero, as shown in Corollary 2.3 (See Fig. 2b for the visualization). As with empirical CTKs, the empirical NTKs are positively skewed, but their eigenvalue scales are much larger than CTKs. In summary, our empirical study demonstrates that although an empirical CTK can have up to NK non-zero eigenvalues, only a few eigenvalues are critical to the scale of $\sum_{i=1}^{P} h(\beta_i)$.

