# OpenReview forum: "Scale-invariant Bayesian Neural Networks with Connectivity Tangent Kernel"
_ICLR.cc/2023/Conference — ICLR 2023 notable top 25%_

### Official Review · Reviewer_RRtQ · 2022-10-19

**Confidence:** 3
**Correctness:** 3
**Technical Novelty And Significance:** 3
**Empirical Novelty And Significance:** 2
**Recommendation:** 6

**Clarity, Quality, Novelty And Reproducibility:**

See the comments above. Clarity and quality have several problems. Given the details provided in the appendix the approach should be reproducible. Unfortunately, my knowledge of the current PAC-Bayes literature is too limited to properly comment on the proposal's novelty, but as far as I can see, novelty is given.

**Strength And Weaknesses:**

## Strengths
The proposed PAC-Bayes bound stays stable and independent wrt rescalings of parameter rescalings compared to the baseline bound. The authors are encouraged to move Table 7 to the main paper as it clearly demonstrates the improvement provided by their approach. Keeping it in the appendix essentially hides this.

A major weakness of the draft is its written structure. There are a lot of grammatical mistakes (primarily missing articles), lots of typos, and partially repeated sentences,... Thorough proofreading is very necessary for any revision.


### Questions
- In the abstract, the authors discuss the flatness hypothesis as being widely accepted, while the first two paragraphs of the introduction then consist of summarizing an ongoing discussion in the literature. Can the authors comment on this?
- Can the authors explain what they mean by the decomposition of scale and connectivity? The connectivity is modified in the Lee et al. paper you cite c is a binary mask, whereas for you it is a continuous vector. In which sense do you get a decomposition between scale and connectivity by the $\theta(1+c)$ formulation? I seem to be misunderstanding a basic point of your setup. E.g., a c=0 via your prior changes nothing in any weight connectivity.
-  Table 1: Why do only some rows contain standard deviations (or standard errors?) and over which number of repetitions are these computed? (Same question for any other table containing stds)
- What is the relation between Table 4 and Figure 1? From my reading, they show the same experimental setting, but if so then the results in the table are misrepresentative as, e.g., for ECE Figure 1 clearly shows lower values for LL than claimed there.

### Further comments
- before eq (2) speaks about "_our_ data-dependent prior" vs _a_ data-dependent prior making it sound as if the split of training data to get a data-dependent prior was a contribution of the current paper.
- $P_c$ appears in the line before (2) without any introduction and then disappears again for the remainder of the paper
- Table 3's formatting makes it look as if CL were to improve also about deep ensembles as well as MC dropout. This needs a major restructuring! (E.g. in the direction of Table 4)

**Summary Of The Paper:**

The authors construct a PAC-Bayes bound that is insensitive to the fact that some parameters of neural nets can be arbitrarily rescaled due to normalization layers rectifying such rescalings. In a second step, they design a variant of the Laplacian approximation and evaluate both on a series of experiments.


**Summary Of The Review:**

A paper with some potential whose main problems are currently in its presentation.

---

> ### Author Response · Authors · 2022-11-15
> **Response**
>
> We appreciate your constructive comments. We give point-to-point replies to your comments in the following.
>
> **Q1**: A major weakness of the draft is its written structure. There are a lot of grammatical mistakes (primarily missing articles), lots of typos, and partially repeated sentences,... Thorough proofreading is very necessary for any revision.
>
> **A1**: Sorry for the confusion. We believe that the revision has removed all such grammatical errors and typos. In addition, all parts that lack clarity have been corrected throughout the paper (Nevertheless, our paper is still a bit dense in some parts since there are several important components to cover in our story. Instead, all of those are added in the appendix in the revision). Note that minor revisions such as simple clarity improvement, grammatical errors, or typo corrections were carried out throughout the paper; they are not colored in the revision. We marked in blue only the parts where the meaning was changed or added.
>
> **Q2** In the abstract, the authors discuss the flatness hypothesis as being widely accepted, while the first two paragraphs of the introduction then consist of summarizing an ongoing discussion in the literature. Can the authors comment on this?
>
> **A2** In the abstract of the revised version, we will avoid confusion with mild expressions such as 'recognized' instead of 'accepted'. While the FM hypothesis is widely recognized as one of the strongest predictors of generalization, there is an issue of choice between several flatness measures. For example, while traces of Hessian and Fisher show the competitive predictive performance of generalization in Table 2, these values have scale-variant issues, as shown in Dinh et al. (2017) and Li et al. (2018). Due to this, ongoing discussions in the literature have primarily focused on developing improved **flatnesses measure** (as a part of generalization bound) without this problem (Tsuzuku et al. (2020) and Kwon et al. (2021)). Taking this discussion further, we present **a generalization bound** without this problem.
>
> **Q3** Can the authors explain what they mean by the decomposition of scale and connectivity? The connectivity is modified in the Lee et al. paper you cite c is a binary mask, whereas for you it is a continuous vector. In which sense do you get a decomposition between scale and connectivity by the θ(1+c) formulation? I seem to be misunderstanding a basic point of your setup. E.g., a c=0 via your prior changes nothing in any weight connectivity.
>
> **A3**: Precisely, our formulation decomposes the scale and connectivity of **perturbations**. We clarified this point in the abstract and Sec. 2.2 in the revised version. Formally, (although the contexts of the two papers are completely different,) the connectivity $c'$ in Lee et al. (2018) is **equivalent to our settings through reparameterization** $c' = c + \mathbf{1_P}$ where $c$ is our connectivity. Note the continuous connectivity is also used by Lee et al. (2018): *"Therefore, by relaxing the binary constraint on the indicator variables $c$, $\Delta L_j$ can be approximated by the derivative of $L$ with respect to $c_j$, which we denote $g_j (\mathbf{w}; \mathcal{D})$."*. Moreover, Lee et al. (2018) measure the sensitivity of loss to connectivity at $c'=\mathbf{1_P}$ in equation (5), which means fully connected NNs without pruning. It's equivalent to our Jacobian for $c=\mathbf{0_P}$, which means that the parameter hasn't been perturbed.

---

> > ### Author Response · Authors · 2022-11-15
> > **Response**
> >
> > **Q4**: Table 1: Why do only some rows contain standard deviations (or standard errors?) and over which number of repetitions are these computed? (Same question for any other table containing stds)
> >
> > **A4**:  In the submitted version, we only provided error bars only for the important quantities: test/train errors for samples of PAC-Bayes posterior.
> >
> > In the revised version, we have included error bars for all rows in Table 1 by evaluating  PAC-Bayes-CTK (eq. (8)) and NTK (eq. (18)) for 3 random seeds. In Table 3, we used 8 random seeds to compute the error bars as mentioned in Sec. 4.2 of the submitted version (or Appendix H of the revised version). In Table 4, We used 4 random seeds to compute the error bars except for Deep Ensemble, which ensembles the NNs from 4 different random seeds. In the revised version, we clarified this information in Appendix H.
> >
> > **Q5**: What is the relation between Table 4 and Figure 1? From my reading, they show the same experimental setting, but if so then the results in the table are misrepresentative as, e.g., for ECE Figure 1 clearly shows lower values for LL than claimed there.
> >
> > **A5**: Your understanding of the relation between Table 4 and Fig. 1 is correct. We apologize for the typos in Table 4. The ECE values of LL/CL in Table 4 should be $0.0532 \pm 0.0010$/$0.0524 \pm 0.0019$, respectively. We have corrected this in the revised version.
> >
> > **Q6**: before eq (2) speaks about "our data-dependent prior" vs a data-dependent prior making it sound as if the split of training data to get a data-dependent prior was a contribution of the current paper.
> >
> > **A6**: Thank you for pointing it out. In the revised version, we corrected this to "a data-dependent prior" to clarify that a data-dependent prior is not a contribution of our paper.
> >
> > **Q7**: Pc appears in the line before (2) without any introduction and then disappears again for the remainder of the paper
> >
> > **A7**: Again, we apologize for the confusion due to typos. $\mathbb{P}_c$ in the line before eq. (2) should be $\mathbb{P}_\mathcal{D}$, which means data-dependent prior. We have corrected this typo in the revised version.
> >
> > **Q8**: Table 3's formatting makes it look as if CL were to improve also about deep ensembles as well as MC dropout. This needs a major restructuring! (E.g. in the direction of Table 4)
> >
> > **A8**: We chose the formatting of Table 3 due to visibility: We found font size would be too small if we transpose Table 3 following the direction of Table 4. We have transposed Table 3 in Appendix K of the revised version. To prevent misunderstanding of CL, we have included a horizontal line between LL and CL in tables.
> >
> > **References**
> >
> > [1] Chaudhari, Pratik, Choromanska, Anna, Soatto, Stefano, Le- Cun, Yann, Baldassi, Carlo, Borgs, Christian, Chayes, Jen- nifer, Sagun, Levent, and Zecchina, Riccardo. Entropy-sgd: Biasing gradient descent into wide valleys. In ICLR’2017, arXiv:1611.01838, 2017.
> >
> > [2] Dinh, Laurent, et al. "Sharp minima can generalize for deep nets." *International Conference on Machine Learning*. PMLR, 2017.
> >
> > [3] Li, Hao, et al. "Visualizing the loss landscape of neural nets." *Advances in neural information processing systems* 31 (2018).
> >
> > [4] Tsuzuku, Yusuke, Issei Sato, and Masashi Sugiyama. "Normalized flat minima: Exploring scale invariant definition of flat minima for neural networks using pac-bayesian analysis." *International Conference on Machine Learning*. PMLR, 2020.
> >
> > [5] Kwon, Jungmin, et al. "Asam: Adaptive sharpness-aware minimization for scale-invariant learning of deep neural networks." *International Conference on Machine Learning*. PMLR, 2021.
> >
> > [6] Lee, Namhoon, Thalaiyasingam Ajanthan, and Philip HS Torr. "Snip: Single-shot network pruning based on connection sensitivity." *arXiv preprint arXiv:1810.02340* (2018).

---

### Official Review · Reviewer_Q1Nm · 2022-10-22

**Confidence:** 3
**Correctness:** 2
**Technical Novelty And Significance:** 3
**Empirical Novelty And Significance:** 2
**Recommendation:** 6

**Clarity, Quality, Novelty And Reproducibility:**

This is well-written, and has good quality. But there are several issues that are unclear to me.

1. I understand the motivation of using Laplace approximation for tractable inference. Nevertheless, is this equivalent to assume that the posterior distribution over parameters is Gaussian? I think the application scope of Laplace approximation needs to be well discussed.

2. The so-called scale-invariance is based on $\theta^* \odot c$ instead of a single parameter $\delta$. Intuitively, the invariance is based on $\theta^* + c \theta^*/ || \theta^*||_2$, which appears feasible to Proposition 2.2. What’s the difference between these two settings?

3. Theorem 2.3 is incomplete as $\sum_{i=1}^P h(\beta_i)$ can be further estimated. More importantly, the results are only valid for the $P \ll N$ case, can not be applied to the commonly-used over-parameterized DNNs due to $\mu_{\mathbb{Q}} \in \mathbb{R}^P$. How to ensure  $|| \mu_Q ||_2^2 \ll N_Q $?

4. In Section 2.4, the experiments work in $|S_P| \gg |S_Q|$. What is the result of the opposite setting? E.g., few training data for $S_P$.


Minor issues:
Table 1 and 7 can be incorporated into one figure for better illustration in the main text.


**Strength And Weaknesses:**

**Pros:**

1. Propose an (empirical) connectivity tagenet kernel (CTK) to demonstrate the sharpness and scaling invariance, which can be regarded an weighted NTK by $\theta^*$ and Jacobian wr.t. $c$
2. Build the PAC-Bayesian bound via CTK for generalization and connect it to invariance for flatness


**Cons:**

There are several points on the motivation of Laplace approximation and the extension to over-parameterization that require more refinement. In my view, this work is built on a certain type of Taylor expansion, obtains the corresponding NTK matrix, assumes the posterior distribution to be Gaussian via Laplace approximation, and derives the PAC-Bayesian bound for under-parameterized deep neural networks. The detailed comments are as below.


**Summary Of The Paper:**

This work centers around the relationship between sharpness (scaling invariance) and generalization in DNNs under the PAC-Bayesian framework. This work decomposes the scale and connectivity parameters, derive the related connectivity tangent kernel (CTK), and builds the PAC-Bayesian generalization bounds. The trace of CTK can be regarded as a good indicator to show the correlation between sharpness and generalization.


**Summary Of The Review:**

This work introduces new ideas/insights to analyze the relationship between sharpness and generalization bounds, but leaves some issues that I am concerned about. I'm hesitant to give a 5 or 6, and would like to see the authors’ rebuttal.

---

> ### Author Response · Authors · 2022-11-15
> **Response**
>
> Thank you for your valuable comments and suggestions! We give point-to-point replies to your questions in the following.
>
> **Q1**: I understand the motivation of using Laplace approximation for tractable inference. Nevertheless, is this equivalent to assuming that the posterior distribution over parameters is Gaussian? I think the application scope of Laplace approximation needs to be well discussed.
>
> **A1**: The Laplace Approximation (LA; eq. 1) was originally proposed for tractable inference of any posterior (including NNs) using **Gaussian distribution**, as mentioned in Bishop et al. (2006): "Here we introduce a simple, but widely used, framework called the Laplace approximation, that aims to find a Gaussian approximation to a probability density defined over a set of continuous variables."
>
> Meanwhile, our PAC-Bayes posterior (eq. 5) is the posterior of Bayesian linear regression whose prior is our data-dependent PAC-Bayes prior (eq. 4), as shown in Appendix D of the revised version. We introduced LA to demonstrate that our PAC-Bayes posterior (eq. 5) is closely related to LA, as shown in Sec. 3.
>
> Please let us know if you were meant to be on other aspects of Laplace Approximation.
>
> **Q2**: The so-called scale-invariance is based on $\theta^* \odot c$ instead of a single parameter $\delta$. Intuitively, the invariance is based on $\theta^* + c \frac{\theta^*}{\lVert \theta^* \rVert}$ which appears feasible to Proposition 2.2. What’s the difference between these two settings?
>
> **A2**: There are two main differences between our perturbation ($\theta^* \odot c$) and the perturbation parameterized as $\theta^* + c \frac{\theta^*}{\lVert \theta^*\rVert}$ where $c \in \mathbb{R}$.
>
> First, the scale-invariance property in Proposition 2.2 does not hold for the perturbation parameterized as $c \frac{\theta^*}{\lVert \theta^* \rVert}$. Instead, this perturbation is only invariant to function-preserving **isotropic** transformations, whose matrix representations are isotropic matrices (i.e., $\lambda I$). Concretely, this is the result of the norm term in the denominator. Therefore, this setting cannot cover activation-wise rescaling transformation in equation (19).
>
> Another main difference between these two parameterizations is the **dimension of the PAC-Bayes posterior distribution** under consideration: The support of PAC-Bayes posterior for our setting is full-dimensional space of non-zero parameters ($\mathbb{R}^P$), while the support of PAC-Bayes posterior for $\theta^* + c \frac{\theta^*}{\lVert \theta^* \rVert}$ is the span of the normalized parameter (i.e., one-dimensional space).
>
> **Q3**: Theorem 2.3 is incomplete as $\Sigma_{i=1}^{P} h(\beta_i)$ can be further estimated. More importantly, the results are only valid for the $P \ll N$ case, can not be applied to the commonly-used over-parameterized DNNs due to $\mu_\mathbb{Q} \in \mathbb{R}^{P}$. How to ensure $\lVert \mu_\mathbb{Q} \rVert^2_2 \ll N_\mathbb{Q}$?
>
> **A3**: Regarding the first question, the existence of $\Sigma_{i=1}^{P} h(\beta_i)$ does not affect to scale-invariance property of PAC-Bayes-CTK as $\Sigma_{i=1}^{P} h(\beta_i)$ is invariant to function-preserving scale transformations: It only depends on the eigenvalues of CTK, which is invariant to parameter scale.
>
> For the second question, although it is hard to guarantee that $\lVert \mu_\mathbb{Q}\rVert^2_2 \ll N_\mathbb{Q}$ holds theoretically, we can expect that $\lVert \mu_\mathbb{Q} \rVert^2_2 \ll N_\mathbb{Q}$ can hold even in many real settings because $\lVert \mu_\mathbb{Q} \rVert$ is not the norm of the **total parameter**, $\lVert \theta^* + \theta^* \odot \mu_\mathbb{Q} \rVert$, but the norm of **perturbation part of the total parameter**. Since the fine-tuning data ($\mathcal{S}_\mathbb{Q}$) is sampled from the same distribution as the pre-trained data ($\mathcal{S}_\mathbb{P}$), the norm of perturbation would not be large in general.
>
> We empirically verified that this holds for ResNet-18, as shown in Table 1. Notably, ResNet-18 has about **200 times** more parameters (i.e., 11M) than the training dataset (CIFAR).  Concretely, the estimated value of $\mu_\mathbb{Q}$ is about 6 for CIFAR-10 and 14 for CIFAR-100 as shown in the "Perturbation" row of Table 1, and the $N_\mathbb{Q}$ is 5,000 for both datasets.

---

> > ### Author Response · Authors · 2022-11-15
> > **Response**
> >
> > **Q4**: In Section 2.4, the experiments work in $|\mathcal{S}_\mathbb{P}|\gg |\mathcal{S}_\mathbb{Q}|$. What is the result of the opposite setting? E.g., few training data for SP.
> >
> > **A4**: Following the reviewer's recommendation, we have conducted additional experiments in Sec. 2.4. with $|\mathcal{S}_\mathbb{P}|=5000$ and $|\mathcal{S}_\mathbb{Q}|=45000$. The following table presents the results for PAC-Bayes-CTK and NTK estimation with this setting.
> >
> > | CIFAR-10 | PAC-Bayes-CTK |  |  |  | PAC-Bayes-NTK |  |  |  |
> > |:---:|:---:|:---:|:---:|:---:|:---:|:---:|:---:|:---:|
> > | Parameter scale | 0.5 | 1.0 | 2.0 | 4.0 | 0.5 | 1.0 | 2.0 | 4.0 |
> > | Trace | 13041 ± 1340 | 13064 ± 1344 | 13057 ± 1344 | 13061 ± 1344 | 5413637 ± 325789 | 1394973 ± 90592 | 545037 ± 6304 | 488604 ± 8954 |
> > | Perturbation | 335.0157 ± 9.5699 | 338.8471 ± 9.3901 | 337.2096 ± 9.4698 | 335.8712 ± 8.3927 | 202.5342 ± 3.1387 | 315.9919 ± 2.7542 | 447.9744 ± 3.7511 | 549.3535 ± 7.6435 |
> > | Sharpness | 13.4824 ± 0.6300 | 13.4927 ± 0.6291 | 13.4908 ± 0.6305 | 13.4941 ± 0.6290 | 64.6953 ± 0.5463 | 52.5450 ± 0.5820 | 44.2475 ± 0.1142 | 42.2973 ± 0.2046 |
> > | KL | 174.2491 ± 5.0337 | 176.1699 ± 4.9428 | 175.3502 ± 4.9896 | 174.6827 ± 4.4582 | 133.6147 ± 1.4843 | 184.2685 ± 1.2386 | 246.1109 ± 1.8574 | 295.8254 ± 3.8179 |
> > | Test err. | 0.2375 ± 0.0017 | 0.2387 ± 0.0019 | 0.2394 ± 0.0022 | 0.2394 ± 0.0022 | 0.3133 ± 0.0030 | 0.2799 ± 0.0020 | 0.2615 ± 0.0022 | 0.2564 ± 0.0020 |
> > | Bound | 0.2771 ± 0.0015 | 0.2782 ± 0.0017 | 0.2784 ± 0.0017 | 0.2785 ± 0.0017 | 0.3359 ± 0.0025 | 0.3059 ± 0.0010 | 0.2966 ± 0.0014 | 0.2998 ± 0.0018 |
> >
> >
> > | CIFAR-100 | PAC-Bayes-CTK |  |  |  | PAC-Bayes-NTK |  |  |  |
> > |:---:|:---:|:---:|:---:|:---:|:---:|:---:|:---:|:---:|
> > | Parameter scale | 0.5 | 1.0 | 2.0 | 4.0 | 0.5 | 1.0 | 2.0 | 4.0 |
> > | Trace | 22168 ± 1431 | 22156 ± 1438 | 22167 ± 1426 | 22166 ± 1431 | 6557644 ± 234835 | 1670224 ± 57783 | 570114 ± 39210 | 505707 ± 47096 |
> > | Perturbation | 447.8628 ± 12.4974 | 447.8570 ± 8.4664 | 445.7568 ± 7.7510 | 446.9493 ± 8.0771 | 399.6581 ± 10.4402 | 622.1746 ± 27.9164 | 790.6263 ± 46.7006 | 823.0565 ± 55.5815 |
> > | Sharpness | 17.1074 ± 0.3992 | 17.1027 ± 0.4024 | 17.1075 ± 0.3973 | 17.1067 ± 0.3986 | 66.3580 ± 0.2726 | 54.1271 ± 0.2521 | 44.4671 ± 0.5871 | 42.6411 ± 0.7865 |
> > | KL | 232.4851 ± 6.1763 | 232.4799 ± 4.1645 | 231.4321 ± 3.7885 | 232.0280 ± 3.9391 | 233.0081 ± 5.2640 | 338.1509 ± 13.9632 | 417.5467 ± 23.0571 | 432.8488 ± 27.4084 |
> > | Test err. | 0.6507 ± 0.0033 | 0.6515 ± 0.0038 | 0.6521 ± 0.0039 | 0.6522 ± 0.0039 | 0.6992 ± 0.0050 | 0.6903 ± 0.0038 | 0.6837 ± 0.0049 | 0.6863 ± 0.0050 |
> > | Bound | 0.7098 ± 0.0035 | 0.7104 ± 0.0038 | 0.7107 ± 0.0038 | 0.7110 ± 0.0039 | 0.7110 ± 0.0027 | 0.6971 ± 0.0059 | 0.7144 ± 0.0062 | 0.7361 ± 0.0049 |
> >
> > These results also have included in Appendix I of the revised version. Similar to the case of $|\mathcal{S}_\mathbb{P}| \ll |\mathcal{S}_\mathbb{Q}|$, PAC-Bayes-CTK shows stable bound for various scales of parameters. On the other hand, PAC-Bayes-NTK is sensitive to parameter scale. Furthermore, we found that non-vacuousness of bounds is still maintained.
> >
> > **Q5**: Minor issues: Table 1 and 7 can be incorporated into one figure for better illustration in the main text.
> >
> > **A5**: Thank you for pointing it out. For better readability, we merged Tables 1 and 7 into a single table in the revised version following the suggestions of the reviewer. We also included error bars for all rows in Table 1 with three random seeds.
> >
> > **References**
> >
> > [1] Bishop, Christopher M., and Nasser M. Nasrabadi. Pattern recognition and machine learning. Vol. 4. No. 4. New York: springer, 2006.

---

> > ### Comment · Reviewer_Q1Nm · 2022-11-15
> > **Laplace approximation and more analysis on $\sum_{i=1}^P h(\beta_i)$**
> >
> > thanks for the authors' response.
> >
> > 1) if my understanding was right, the Laplace approximation requires the posterior distribution to be Gaussian. It's ok to me but it would be better to add some discussion or analysis on error estimation when introducing an approximation scheme.
> >
> > 2) $\sum_{i=1}^P h(\beta_i)$ requires more analysis based on some certain eigenvalue decay of CTK.
> >
> > minor issue:
> > in Table 1, the results can be rounded to first or second digit for better illustration as current font size is small due to the margin limit.

---

> > > ### Author Response · Authors · 2022-11-17
> > > **Response**
> > >
> > > We appreciate your additional comments and suggestions. We give point-to-point replies in the following.
> > >
> > > **Q1**: if my understanding was right, the Laplace approximation requires the posterior distribution to be Gaussian. It’s ok to me but it would be better to add some discussion or analysis on error estimation when introducing an approximation scheme.
> > >
> > > **A1**: Thanks for the useful suggestion. Again it is true that the Laplace Approximation (LA) approximates the complex (non-Gaussian) posterior distribution to a Gaussian distribution. While estimation error may occur when approximating a non-Gaussian posterior to Gaussian, most recent studies on LA  (Ritter et al., 2018; Kristiadi et al., 2020; Daxberger et al., 2021b) for NNs do not analyze the estimation error. This is due to the difficulty of analytically expressing the exact posterior of Bayesian NNs. Furthermore, there is no estimation error in PAC-Bayes-CTK (Theorem 2.2), since our PAC-Bayes-CTK is a generalization bound for our PAC-Bayes posterior (equation 5) rather than a generalization bound for the exact posterior of NNs.
> > >
> > > As a follow-up study, we expect that the effectiveness of generalizability and uncertainty can be evaluated by assuming prior and posterior distributions for other distributions beyond the Gaussian distribution. Reflecting on the reviewer's opinion, we add it to the conclusion.
> > >
> > > **Q2**: $\sum_{i=1}^{P} h(\beta_i)$ requires more analysis based on some certain eigenvalue decay of CTK.
> > >
> > > **A2**: Thanks for the useful suggestion. Regarding this, please reconsider that we performed an eigenvalue-based analysis of $\sum_{i=1}^{P} h(\beta_i)$ in Corollary 2.3: As we showed in the proof of Corollary 2.3, the value of the term $h(\beta_i)$ equals zero when the $i$-th eigenvalue of CTK becomes zero. Therefore, many terms in $\sum_{i=1}^{P} h(\beta_i)$ decay to zero as $NK$ is usually much smaller than the number of parameters ($P$).
> > >
> > > To analyze the remaining non-zero eigenvalues, we conducted an empirical study in Appendix L. Fig. 3 shows that empirical CTKs have positively skewed (i.e., right-tailed) eigenspectrums with modes close to 0. In other words, many of the non-zero eigenvalues of CTK are also close to zero. Therefore, the corresponding $h(\beta_i)$ for these eigenvalues are also close to zero, as shown in Corollary 2.3 (See Fig. 2 (b) for the visualization). As with empirical CTKs, the empirical NTKs are positively skewed, but their eigenvalue scales are much larger than CTKs.
> > >
> > > In summary, our empirical study demonstrates that $\sum_{i=1}^{P} h(\beta_i)$ term becomes $\sum_{i=1}^{T} h(\beta_i )$ , where $T (<NK<P)$ is the number of non-zero eigenvalues of CTK. This means that the proposed bound does not depend on the parameter scale $P$, but rather on the tighter information value, $T$ (rank of nonzero eigenspace for our CTK).
> > >
> > > **Q3**: minor issue: in Table 1, the results can be rounded to first or second digit for better illustration as current font size is small due to the margin limit.
> > >
> > > **A3**: Thank you for pointing it out! Following the reviewer's suggestion, we round all results in Tables 1 and 7 to the second digit. Furthermore, we multiplied the power of tens for the results of Trace ($10^{-4}$), Test error ($10^{2}$), and Bound ($10^{2}$) rows for better font size and illustration.
> > >
> > > **References**
> > >
> > > [1] Ritter, Hippolyt, Aleksandar Botev, and David Barber. "A scalable laplace approximation for neural networks." *6th International Conference on Learning Representations, ICLR 2018-Conference Track Proceedings*. Vol. 6. International Conference on Representation Learning, 2018.
> > >
> > > [2] Kristiadi, Agustinus, Matthias Hein, and Philipp Hennig. "Being bayesian, even just a bit, fixes overconfidence in relu networks." *International conference on machine learning*. PMLR, 2020.
> > >
> > > [3] Daxberger, Erik, et al. "Bayesian deep learning via subnetwork inference." *International Conference on Machine Learning*. PMLR, 2021.

---

> > > > ### Comment · Reviewer_Q1Nm · 2022-11-17
> > > > **increase score to 6**
> > > >
> > > > Thanks for the authors' update.
> > > >
> > > > I increase my score to 6.
> > > > Actually it would be better to study the eigenvalue decay of CTK kernel, maybe polynomial decay, similar to [1,2], but it might be a bit beyond the scope of this work.
> > > >
> > > > [1] Geifman, Amnon, Abhay Yadav, Yoni Kasten, Meirav Galun, David Jacobs, and Basri Ronen. "On the similarity between the laplace and neural tangent kernels." NeurIPS 2020.
> > > >
> > > > [2] Chen, L. and Xu, S., 2020. Deep neural tangent kernel and laplace kernel have the same RKHS. ICLR2021.

---

> > > > > ### Author Response · Authors · 2022-11-17
> > > > > **Thanks for the discussion and raising the score**
> > > > >
> > > > > Dear Reviewer Q1Nm,
> > > > >
> > > > > We deeply appreciate the discussion and increasing the score of our paper.
> > > > >
> > > > > Best regards,
> > > > > Authors of paper 5870

---

### Official Review · Reviewer_HGSe · 2022-10-24

**Confidence:** 4
**Correctness:** 4
**Technical Novelty And Significance:** 3
**Empirical Novelty And Significance:** 3
**Recommendation:** 8

**Clarity, Quality, Novelty And Reproducibility:**

**Clarity**

I found this paper difficult to read due to the abundance of grammatical errors. I would recommend the authors employ a free online grammar checker.

The issue of scale-invariance in neural networks with normalisation layers was only discussed in the introduction and at a very high level. I would recommend the authors provide a more concrete description of this problem, which is a central motivation of the proposed techniques.

**Novelty**

The issue of scale-invariance in NNs with normalisation layers is known within both the optimisation community (https://arxiv.org/abs/1706.05350) and the Bayesian deep learning community (https://arxiv.org/pdf/2206.08900.pdf). However, to the best of my knowledge, the solution proposed in this paper is novel.

To the best of my knowledge, the PAC-Bayes bound used by the authors, where a NN is trained and a linear model is built on top of its Jacobian features, is novel.

**Reproducibility**

The authors do not provide a code release or a very detailed experimental setup description. Given that I have some experience in this field, I think that reproducing this work would be challenging but not impossible.

**Strength And Weaknesses:**

**Strengths**

* This paper makes 2 technically strong contributions, each of which could almost be a paper on its own.
	* A new non-vacuous PAC-based bound for a linear model built on a basis chosen to be the Jacobian of a pre-trained NN.
	* A formulation of the above linear model that avoids pathologies related to scale invariance in neural networks

* The experimental validation is also quite strong. I was impressed that the authors evaluated both their generalisation bound and uncertainty estimation methods using ResNet18 (an 11M parameter model) on the CIFAR100 dataset.


**Weaknesses**

* Motivation / Problem exposition. The scale-variance issues this paper addresses are quite subtle and are not explained in enough detail. I was lucky to be familiar with these issues as I have dealt with them in my own work. However, I think that the average reader will be quite confused about what problem the authors are solving. I would recommend including a specific example of the Jacobians of a NN changing when the scale of the weights changes, despite the NN output not changing at all.

* The authors evaluate their bound and show it gives very similar results for different scalings of the NN parameters. However, no other non-invariant bounds are shown. I think that it would be illustrative for the reader to see how the tightness of alternative bounds is lost due to the invariance pathologies addressed by this paper.

* A potential issue with the proposed approach is that the re-scaling of the Jacobian basis that is proposed depends on the linearisation point. This vector contains information about the (potentially arbitrary) scaling of the dimensions of the NN Jacobian and information about how to fit the data well with the NN. It is not clear that the latter will be helpful to scale the Jacobian basis and it could in fact introduce some bias into the linearised model.


* I found the experimental details provided lacking, even after reading the appendix. In this regard, I found the ResNet-18 CIFAR100 experiments particularly surprising. The dimensionality of the Jacobian basis expansion for a single input point is NOutputs x Nweights and evaluating this expansion requires as many backward passes as output dimensions. Additionally, a single step of Lanczos iteration requires a pass through the whole dataset and asymptotically the method requires as many steps as the rank of the kernel matrix being decomposed. How did the authors deal with these challenges?

* I would suggest the authors repeat the evaluations from tables 1 and 2 for multiple seeds and add errorbars.

* I found the clarity of writing to be poor, partly due to the text being full of grammatical mistakes.  I provide more details below.

**Summary Of The Paper:**

This paper tackles the characterisation of the local curvature of NN loss landscapes in the face of scale-symmetries that appear in these models’ loss landscapes. Specifically, the authors build upon recent work in the linearised Laplace method for uncertainty estimation with neural networks by extending it to account for scale invariance in the NN parameter space.

The authors first use their construction, dubbed the connectivity tangent Kernel (CTK), to build a PAC-Bayes bound with a data-dependent prior that is invariant to the scale of the NN weights. The authors show this bound to be non-vacuous empirically on CIFAR10 and CIFAR100 with ResNet-18. The authors then propose the trace of the  CTK as an easy-to-compute surrogate quantity for estimating models’ generalisation. They show this quantity to correlate more strongly with test performance than previously proposed quantities, such as the trace of the Fisher information matrix.

Finally, the authors use their connectivity tangent kernel for uncertainty estimation within the linearised Laplace framework. They apply their approach to UCI regression datasets and to CIFAR10+100 with ResNet18. The proposed method provides a modest but significant performance boost on top of the standard linearised Laplace approach.

**Summary Of The Review:**


I think this is a technically strong paper which I would like to see accepted. The paper has some major issues regarding clarity but I am confident that the authors can address these in a revision. When they do, I will be happy to raise my score.

---

> ### Author Response · Authors · 2022-11-15
> **Response**
>
> We appreciate your constructive feedback! We give point-to-point replies to your questions in the following.
>
> **Q1**: Motivation / Problem exposition. The scale-variance issues this paper addresses are quite subtle and are not explained in enough detail. I was lucky to be familiar with these issues as I have dealt with them in my own work. However, I think that the average reader will be quite confused about what problem the authors are solving. I would recommend including a specific example of the Jacobians of a NN changing when the scale of the weights changes, despite the NN output not changing at all.
>
> **A1**: In Appendix D of the submitted version (or Appendix E of the revised version), we discussed representative cases of function-preserving scale transformations. These include activation-wise rescaling transformations (Neyshabur et al., 2015), which multiplies the same scaling factor in one layer and divides it into the next layer, and weight decay (WD) with BN in Zhang et al. (2019). We refer to Appendix D in Sec. 2.2 where function preserving scale transformation first appears.
>
> To help unfamiliar readers understand, we also have included toy examples to clarify the underlying computations of activation-wise rescaling and WD with BN in Appendix E of the revised version.
>
> Please let us know if you were meant to be on other aspects of the scale-variance issue.
>
> **Q2**: The authors evaluate their bound and show it gives very similar results for different scalings of the NN parameters. However, no other non-invariant bounds are shown. I think that it would be illustrative for the reader to see how the tightness of alternative bounds is lost due to the invariance pathologies addressed by this paper.
>
> **A2**: Theoretically, recent bounds of PAC-Bayes analysis are not invariant to scale transformations due to the scale-dependent term (equation (34) in Tsuzuku et al. (2020) and equation (6) in Kwon et al. (2021)), as mentioned in Sec. 2.3. Concretely, these terms are proportional to the norm of pre-trained parameters. We have clarified this in Sec. 2.3 of the revised version.
>
> In our paper, we demonstrated the PAC-Bayes-NTK bound (Theorem B.1), a scale-variant counterpart of PAC-Bayes-CTK. Empirically, we evaluated the PAC-Bayes-NTK bound in Table 7 of the submitted version. In this empirical study, it is demonstrated that PAC-Bayes-NTK is very sensitive to parameter scale. For better readability, we merged Tables 1 and 7 into a single table in the revised version following the suggestions of the reviewers (Q1Nm and RRtQ). We also included error bars for all rows in Table 1 with three random seeds.
>
> **Q3**: A potential issue with the proposed approach is that the re-scaling of the Jacobian basis that is proposed depends on the linearization point. This vector contains information about the (potentially arbitrary) scaling of the dimensions of the NN Jacobian and information about how to fit the data well with the NN. It is not clear that the latter will be helpful to scale the Jacobian basis and it could in fact introduce some bias into the linearised model.
>
> **A3**: It is notable that the ordinary linearization w.r.t. parameter ( $f_{\theta^*}^\mathrm{lin}(x,\delta) = f(x,\theta^*) + \mathbf{J_\theta}(x,\theta^*)\delta$ ) also depends on the linearization point. Therefore, this linearization also can be affected by both scale of parameters and the information about how to fit the data well with the NN. On the other hand, our linearization w.r.t. connectivity (eq. 3) is not influenced by the scale of parameters. As such, our work can be viewed as reducing the bias in existing linearization w.r.t. parameters.
>
> Furthermore, even though we discussed linearized NNs at the pre-trained parameter throughout the paper, the scale-invariance properties (e.g., Proposition 2.2., Theorem 2.3, and Proposition 2.5) of our framework hold for **arbitrary linearization points**: $\mathbf{J_\theta}(x, \mathcal{T}(\psi)) diag (\mathcal{T}(\psi)) = \mathbf{J_\theta}(x, \psi) \mathcal{T}^{-1} \mathcal{T} diag (\psi) = \mathbf{J_\theta}(x, \psi) diag (\psi)$ holds for arbitrary $\psi \in \mathbb{R}^{P}$, as shown in equation (12). Thus, the bias from how to fit the data well can be controlled by choosing linearization points from multiple candidates.
>
> Please let us know if your remarks pertain to other aspects of linearization.

---

> > ### Author Response · Authors · 2022-11-15
> > **Response**
> >
> > **Q4**: I found the experimental details provided lacking, even after reading the appendix. In this regard, I found the ResNet-18 CIFAR100 experiments particularly surprising. The dimensionality of the Jacobian basis expansion for a single input point is NOutputs x Nweights and evaluating this expansion requires as many backward passes as output dimensions. Additionally, a single step of Lanczos iteration requires a pass through the whole dataset and asymptotically the method requires as many steps as the rank of the kernel matrix being decomposed. How did the authors deal with these challenges?
> >
> > **A4**: Our paper's three main implementation targets are Connectivity Laplace, Connectivity Sharpness, and Lanczos iteration for PAC-Bayes-CTK/NTK. To provide details on implementations, we have included **pseudo-codes** and **time and memory complexities** for Connectivity Laplace and Hutchison’s method for Connectivity sharpness in Appendix F. For the Lanczos iteration, we slightly modified the open-sourced implementation of Ghorbani et al. (2019) in https://github.com/google/spectral-density/blob/master/jax/lanczos.py. All related **python snippets** have been included as supplementary materials in the revised version, and the extended version will be uploaded to GitHub.
> >
> > Note that while computing full Jacobians requires repeating multiple backward passes as many times as the output dimension (as pointed out by the reviewer), we used mini-batch Jacobian-vector products (JVPs). It was shown that the time and memory complexities of JVPs are comparable to mini-batch forward propagations by Novak et al. (2022). For Lanczos iterations, a fixed number (e.g., 100) of Lanczos iterations is used instead of decomposing the kernel matrix based on the rank. In terms of wall clock time, it takes about 50 minutes to compute 100 Lanczos iterations, similar to the report of Ghorbani et al. (2019): "For a CIFAR-10, on 10 Tesla P100 GPUs, it takes about an hour to compute 90 Lanczos iterations." This wall time clock is derived from ResNet-18 and CIFAR datasets using eight NVIDIA RTX 3090 GPUs. (For comparison, it took about 20 minutes for pre-training.)
> >
> > **Q5**: I would suggest the authors repeat the evaluations from tables 1 and 2 for multiple seeds and add errorbars.
> >
> > **A5**: Following the reviewer's suggestion, we have included error bars for all rows in Table 1 by evaluating  PAC-Bayes-CTK (eq. (8)) and NTK (eq. (18)) for 3 random seeds in the revised version. However, we emphasize that Table 2 presents the rank correlations computed over 243 models, as mentioned in Appendix H. Therefore, the results in Table 2 are as reliable as the results of multiple repetitions. Also, repeating experiments in Table 2 is  computationally too much of a burden for us. We found that other works also provided results without replicates (e.g., Table 3 in Jiang et al. (2020) and Table 1 in Kwon et al. (2021)).
> >
> > **Q6**: I found this paper difficult to read due to the abundance of grammatical errors. I would recommend the authors employ a free online grammar checker.
> >
> > **A6**: Sorry for the confusion. We believe that the revision has removed all such grammatical errors and typos. In addition, all parts that lack clarity have been corrected throughout the paper (Nevertheless, our paper is still a bit dense in some parts since there are several important components to cover in our story. Instead, all of those are added in the appendix in the revision). Note that minor revisions such as simple clarity improvement, grammatical errors, or typo corrections were carried out throughout the paper; they are not colored in the revision. We marked in blue only the parts where the meaning was changed or added.
> >
> > **References**
> >
> > [1] Neyshabur, Behnam, Russ R. Salakhutdinov, and Nati Srebro. "Path-sgd: Path-normalized optimization in deep neural networks." *Advances in neural information processing systems* 28 (2015).
> >
> > [2] Zhang, Guodong, et al. "Three mechanisms of weight decay regularization." *arXiv preprint arXiv:1810.12281* (2018).
> >
> > [3] Novak, Roman, Jascha Sohl-Dickstein, and Samuel S. Schoenholz. "Fast finite width neural tangent kernel." *International Conference on Machine Learning*. PMLR, 2022.
> >
> > [4] Ghorbani, Behrooz, Shankar Krishnan, and Ying Xiao. "An investigation into neural net optimization via hessian eigenvalue density." *International Conference on Machine Learning*. PMLR, 2019.
> >
> > [5] Jiang, Yiding, et al. "Fantastic generalization measures and where to find them." *arXiv preprint arXiv:1912.02178* (2019).
> >
> > [6] Kwon, Jungmin, et al. "Asam: Adaptive sharpness-aware minimization for scale-invariant learning of deep neural networks." *International Conference on Machine Learning*. PMLR, 2021.

---

> > > ### Comment · Reviewer_HGSe · 2022-11-17
> > > **Thanks for the extensive reply.**
> > >
> > > Thanks for the detailed response. After having a skim of the updated draft, I have increased my score to an 8.

---

> > > > ### Author Response · Authors · 2022-11-17
> > > > **Thanks for the comments and increasing the score**
> > > >
> > > > Dear Reviewer HGSe,
> > > >
> > > > We sincerely appreciate the comments and raising the score of our paper.
> > > >
> > > > Best regards, Authors of paper 5870

---

### Official Review · Reviewer_xWGj · 2022-10-24

**Confidence:** 3
**Correctness:** 3
**Technical Novelty And Significance:** 3
**Empirical Novelty And Significance:** 3
**Recommendation:** 6

**Clarity, Quality, Novelty And Reproducibility:**

**Clarity** Good in terms of the overall flow and structure, but quite underwhelming when it comes to the details. There is just too much notation that is introduced inline throughout the paper, without any aids for the reader, such as a notation table or an algorithm box with pseudocode that recaps things concisely.

**Quality** Overall I'm willing to give the paper the benefit of the doubt and assume that it is good work with interesting empirical and theoretical contributions (although I will caveat this with saying that I did not check the proofs on the PAC Bayes bounds).

**Novelty** The ideas in this work are new.

**Reproducibility** Hyperparameters are vaguely described as a table of ranges in the appendix, however I did not find any final values. Given the further lack of clarity on the method itself, I doubt I would be able to reproduce any of the results in the paper.

**Strength And Weaknesses:**

Strengths:
* The range of contributions is quite broad, from a PAC Bayes bound to a practical approximate inference method.
* The paper is overall well-structured.
* The bounds seem to be empirically reasonably tight.
* The uncertainty estimation method performs competitively.

Weaknesses:
* I found it quite difficult to follow what the paper is doing in detail and would not be confident that I would be able to implement the paper just based on the manuscript. In particular, I don't really understand which dataset split is used for calculating which quantities.
* There are a lot of grammatical mistakes in the manuscript (as far as I can tell as a non-native English speaker), to the extent that it made the paper difficult to read for me (I'd usually just mention this as a minor comment, but in this instance I find that it is to a degree that it negatively impacts the already lacking clarity of the paper).
* There is no discussion of computational cost. It seems like the method needs to evaluate a lot (100 Lanczos iteration; inside of an optimization problem) of Jacobian-vector products, which is not cheap (especially in case each product requires a pass over the training set, although I'm not sure if this is the case).

**************************************************
POST REBUTTAL UPDATE

The authors have extensively updated the paper, addressing the concerns around presentation. I therefore increase my score.

**Summary Of The Paper:**

The paper proposes scaling-invariant prior and posterior distributions for PAC Bayes generalization bounds. It supports these empirically and, based on these, introduces a practical uncertainty estimation method that compares competitively with the linearized Laplace approximation.

**Summary Of The Review:**

An in principle interesting paper with glaring weaknesses in its presentation, which all things considered make me lean towards a rejection.

---

> ### Author Response · Authors · 2022-11-15
> **Response**
>
> Thank you for your constructive comments! We give point-to-point replies to your questions in the following.
>
> **Q1**: I found it quite difficult to follow what the paper is doing in detail and would not be confident that I would be able to implement the paper just based on the manuscript. In particular, I don't really understand which dataset split is used for calculating which quantities.
>
> **A1**: Our paper's three main implementation targets are Connectivity Laplace, Connectivity Sharpness, and Lanczos iteration for PAC-Bayes-CTK/NTK. To provide details on implementations, we included **pseudo-codes** for Connectivity Laplace and Hutchison’s method for Connectivity sharpness in Appendix F. For the Lanczos iteration, we slightly modified the open-sourced implementation of Ghorbani et al. (2019) in https://github.com/google/spectral-density/blob/master/jax/lanczos.py. All related **python snippets** have been included as supplementary materials in the revised version, and the extended version will be uploaded to GitHub.
>
> Regarding the dataset split, we used prior dataset $\mathcal{S}_\mathbb{P}$ for pre-training ($\theta^*$), entire training dataset $\mathcal{S}$ for generating PAC-Bayes posterior, and posterior dataset $\mathcal{S}_\mathbb{Q}$ for evaluating PAC-Bayes-CTK and NTK. In the revised version, we have clarified the usage of the dataset split in the last paragraph of Sec. 2.1: "Following the recent discussion in Perez-Ortiz et al. (2021), ~."
>
> **Q2**: There are a lot of grammatical mistakes in the manuscript (as far as I can tell as a non-native English speaker), to the extent that it made the paper difficult to read for me (I'd usually just mention this as a minor comment, but in this instance I find that it is to a degree that it negatively impacts the already lacking clarity of the paper).
>
> **A2**: Sorry for the confusion. We believe that the revision has removed all such grammatical errors and typos. In addition, all parts that lack clarity have been corrected throughout the paper (Nevertheless, our paper is still a bit dense in some parts since there are several important components to cover in our story. Instead, all of those are added in the appendix in the revision). Note that minor revisions such as simple clarity improvement, grammatical errors, or typo corrections were carried out throughout the paper; they are not colored in the revision. We marked in blue only the parts where the meaning was changed or added.
>
> **Q3**: There is no discussion of computational cost. It seems like the method needs to evaluate a lot (100 Lanczos iteration; inside of an optimization problem) of Jacobian-vector products, which is not cheap (especially in case each product requires a pass over the training set, although I'm not sure if this is the case).
>
> **A3**: In addition to pseudo-codes for convenience of implementation, we have included time and memory complexities in Appendix F.
>
> For Lanczos iteration to compute bounds, we used both mini-batch Jacobian-vector products (JVPs) and vector-Jacobian products (VJPs). For Connectivity Sharpness, we only used mini-batch VJPs. For the inference of LL/CL, we only used mini-batch JVPs.
>
> The time and memory complexities of computing JVPs/VJPs for mini-batch are comparable to forward/backward propagations, as shown in Novak et al. (2022), respectively. In terms of wall clock time, it takes about 5 minutes to sample 8 NNs for LL and CL, takes 1 minute to compute the MC approximation of CS with sample size one, and takes 50 minutes to compute 100 Lanczos iterations. The wall clock times are derived from ResNet-18 and CIFAR datasets using eight NVIDIA RTX 3090 GPUs. (For comparison, it took about 20 minutes for pre-training.) Note that the wall clock time of Lanczos iterations is similar to the report of Ghorbani et al. (2019): "For a CIFAR-10, on 10 Tesla P100 GPUs, it takes about an hour to compute 90 Lanczos iterations."

---

> > ### Author Response · Authors · 2022-11-15
> > **Response**
> >
> > **Q4**: Clarity: Good in terms of the overall flow and structure, but quite underwhelming when it comes to the details. There is just too much notation that is introduced inline throughout the paper, without any aids for the reader, such as a notation table or an algorithm box with pseudocode that recaps things concisely.
> >
> > **A4**: We apologize for the complex inline equations; due to space constraints, we had to include inline equations. To provide details on implementations, we have included **pseudo-codes** in Appendix F, as mentioned earlier. In Appendix A of the revision, we have included a **notation table** for mathematical terms used in the main paper. As this table clarifies the definition and shape of each term, we believe this table will help readers' convenience.
> >
> > **Q5**: Reproducibility: Hyperparameters are vaguely described as a table of ranges in the appendix, however I did not find any final values. Given the further lack of clarity on the method itself, I doubt I would be able to reproduce any of the results in the paper.
> >
> > **A5**: Our paper has three main experiments (computing bounds (Sec. 2.4), UCI regression, and CIFAR classification). In the revision, we have summarized the detailed settings for these three experiments, including hyperparameter selection in Appendix XXX for easy reference.
> >
> > For bound computing experiments, we used the final hyperparameter values as $\delta=0.1, \alpha=0.1, \sigma=1.0$. The values were chosen so that the PAC-Bayes-CTK and NTK bounds are not vacuous.
> >
> > In UCI regression tasks, we fixed $\sigma$ to 1 but searched $\alpha$ as the optimal $\alpha$ varies for each regression dataset.
> >
> > In CIFAR classification tasks, we fixed $\sigma$ to 0.01 and searched $\alpha$ with NLL of the posterior dataset, which is not used for pre-training (See Fig. 1 for the results). The optimal $\alpha$ is 0.01 for LL and 1.0 for CL.
> >
> > **References**
> >
> > [1] Ghorbani, Behrooz, Shankar Krishnan, and Ying Xiao. "An investigation into neural net optimization via hessian eigenvalue density." *International Conference on Machine Learning*. PMLR, 2019.
> >
> > [2] Novak, Roman, Jascha Sohl-Dickstein, and Samuel S. Schoenholz. "Fast finite width neural tangent kernel." International Conference on Machine Learning. PMLR, 2022.
> >
> > [3] James Bradbury, Roy Frostig, Peter Hawkins, Matthew James Johnson, Chris Leary, Dougal Maclaurin, George Necula, Adam Paszke, Jake VanderPlas, Skye Wanderman-Milne, and Qiao Zhang. JAX: composable transformations of Python+NumPy programs. github, 2018.

---

> > > ### Comment · Reviewer_xWGj · 2022-11-18
> > > **Thank you**
> > >
> > > Thank you for the extensive responses to all reviews as well as your updates of the paper. I've not had the chance to re-read it in detail, but based on a superficial pass do believe that the added structure will make it significantly easier to understand things in detail, addressing my core concern. I will therefore raise my score.

---

> > > > ### Author Response · Authors · 2022-11-18
> > > > **Thanks for the suggestions and increasing the score**
> > > >
> > > > Dear Reviewer xWGj,
> > > >
> > > > We truly appreciate your suggestions and raising the score of our paper.
> > > >
> > > > Best regards, Authors of paper 5870

---

### Decision · Program_Chairs · 2023-01-20

**Decision:**

Accept: notable-top-25%

**Justification For Why Not Higher Score:**

Theoretical contributions are great, however, the developed algorithm following the theoretical insights doesn't seem to show significant advantages yet. Also the current presentation is still a bit difficult to read and understand.

**Justification For Why Not Lower Score:**

I think the new concept of connectivity tangent kernel and the new theoretical results with non-vacuous bounds are great to be recognised by the Bayesian Deep Learning community and beyond.

**Metareview: Summary, Strengths And Weaknesses:**

The paper studies the local curvature of NN loss landscapes in the existence of scale-symmetries. The main contributions are on the theoretical side, i.e., the paper proposes the concept of connectivity tangent kernel (CTK) to build a non-vacuous PAC-Bayes bound with a data-dependent and scale-invariant prior. The theory is validated by empirical computation of the bounds vs practical errors using an accompanied Linear Laplace algorithm, and such algorithm is also tested on some Bayesian NN benchmarks to show practical usefulness.

Reviewers all agree that the theoretical contributions of the paper are novel and significant, for which I agree. The provided bound is non-vacuous, and the insights there also lead to a practical algorithm for BNNs.

The main concerns within reviewers are the clarity of presentation (beyond grammar issues) as well as unclear descriptions on the algorithm's implementation details.

I recommend the authors to go for a careful revision process for the final camera ready. The paper's presentation was improved during the discussion period, but still it needs edits to achieve better communication of the results.

**Note From Pc:**

if the above contains the word "oral" or "spotlight" please see: "oral" presentation means -> notable-top-5% and "spotlight" means -> notable-top-25%. As stated in our emails, we are disassociating presentation type from AC recommendations

**Summary Of Ac-Reviewer Meeting:**

N/A